# Maximizing Mutual Information Between Prompt and Response Improves LLM Performance With No Additional Data

**Hyunji Nam** [1]   **Haoran Li** [2]   **Natasha Jaques** [3]

## Abstract

While post-training has successfully improved large language models (LLMs) across a variety of domains, these gains heavily rely on human-labeled data or external verifiers. Existing data has already been exploited, and new data is expensive to collect. Moreover, true intelligence goes far beyond verifiable tasks. Therefore, we need self-improvement frameworks that are less dependent on external signals and more broadly applicable to both verifiable and non-verifiable domains. We propose **Mutual Information Preference Optimization (MIPO)**, a contrastive data augmentation method that constructs preference pairs by generating a positive response conditioning on the correct prompt, and a negative response by conditioning on a random, unrelated prompt. We show that using Direct Preference Optimization to learn from this paired data maximizes pointwise mutual information *under the base LLM* between prompts and model responses. Experiments with with 1-7B parameter Llama and Qwen instruct models show that MIPO achieves 3–16% gains (and 51% increase for Qwen2.5-1.5B-Instruct) on personalization compared to prompting baselines. Surprisingly, MIPO can also be useful in verifiable domains, such as math and multiple-choice question answering, yielding 1–20% gains *without any additional data or external supervision*. These results suggest a promising direction for self-improvement using intrinsic signals derived from contrastive data pairs.

## 1. Introduction

*"Data is the fossil fuel of AI."* – Ilya Sutskever

---

[1]Stanford University [2]Character AI [3]University of Washington. Correspondence to: Hyunji Nam <hjnam@stanford.edu>, Natasha Jaques <nj@cs.washington.edu>.

*Proceedings of the 43$^{rd}$ International Conference on Machine Learning*, Seoul, South Korea. PMLR 306, 2026. Copyright 2026 by the author(s).

Large language models (LLMs) have achieved remarkable success across diverse tasks and domains, from open-ended text generation to reasoning and mathematics. Post-training has been pivotal in driving this success through methods such as Reinforcement Learning with Human Feedback (RLHF) (Stiennon et al., 2022; Touvron et al., 2023; Ouyang et al., 2022) and Reinforcement Learning with Verifiable Rewards (RLVR) (Lambert et al., 2025; Guo et al., 2025). However, these methods still rely heavily on human feedback or external verifiers. As models become increasingly advanced and continue to develop beyond (average) human capabilities, this poses two challenges to existing frameworks: (1) intelligence extends far beyond verifiable tasks, and (2) high-quality human data is expensive to collect. The analogy of data as the "fossil fuel" of AI applies not only to pre-training but also to post-training.

While alternative approaches, such as Reinforcement Learning from AI Feedback (RLAIF) (Bai et al., 2022a; Lee et al., 2024a; Chen et al., 2024a), have been successful for reducing human supervision, self-training *using the same model* without reliance on larger models remains under-explored. In fact, as a potential negative result, Huang et al. (2024) observe that self-corrections by models can degrade performance when external feedback or verifiers are unavailable.

The lack of existing solutions for self improvement motivates our research: ***Can models improve without external signals?*** Existing work shows some promise in this direction: even when the chosen responses are suboptimal, as long as they are better than the rejected counterparts (e.g., generated by a larger model), they can still lead to learning (Yao et al., 2024; Geng et al., 2025). However, they still require a larger model to create supervised labels. If we think back to the literature on computer vision (for example, Krizhevsky et al. (2012)), data augmentation approaches like translation and rotation improved learning without the use of any external supervision. Can we develop an analogous approach for LLMs, creating reliable learning signals intrinsic to the problem, without relying on human data or verifiers?

We propose **Mutual Information Preference Optimization (MIPO)**, a data-augmentation method that maximizes mutual information between prompt $x$ and model response $y$

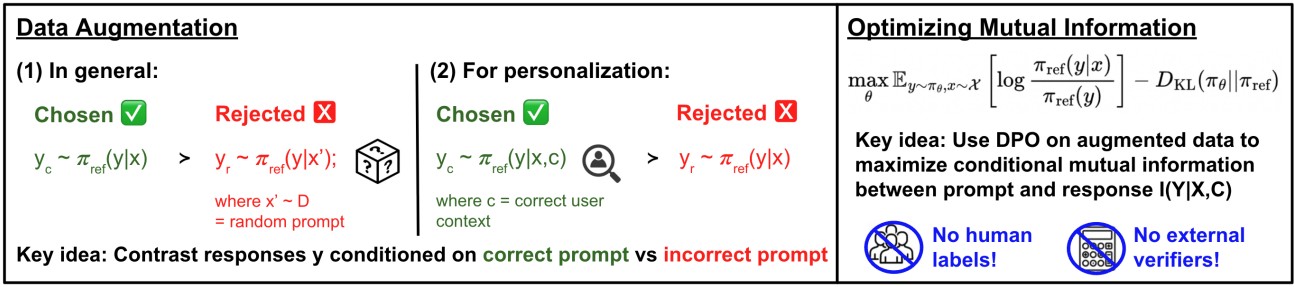

*Figure 1.* We propose an intrinsically motivated post-training method based on mutual information that does not require human labels or verifiers. We present two versions of MIPO: **(1) maximizing mutual information (under the base LLM) between responses and prompts**, and **(2) maximizing conditional mutual information between responses and user contexts given prompts** The second objective is particularly useful for personalization, as it encourages the policy to generate responses that are more likely conditioned on the specific user context, but rare globally.

under the reference LLM. We show that the implicit reward in Direct Preference Optimization (Rafailov et al., 2024) can be cast as the InfoNCE (van den Oord et al., 2019) objective, and we train the policy using chosen responses generated with the correct prompts $y_{\text{chosen}} \sim \pi_{\text{ref}}(y|x)$, and rejected responses generated with random, misspecified prompts $y_{\text{rejected}} \sim \pi_{\text{ref}}(y|x')$ for $x' \neq x$.

We apply this contrastive data technique to personalization problems, where models are given a prompt $x$ along with user-specific instructions or preferences $c$ to guide the personalized response. Context-driven personalization is increasingly becoming important given the growing need for pluralistic alignment (Sorensen et al., 2024). Despite the growing capabilities of models, Jiang et al. (2025b) alarmingly show the risks of model homogeneity, potentially ignoring unique user contexts and circumstances. We apply MIPO to maximize the relative likelihood of generating personalized responses $y_{\text{chosen}} \sim \pi_{\text{ref}}(y|x,c)$ compared to the rejected (un-personalized) responses $y_{\text{rejected}} \sim \pi_{\text{ref}}(y|x)$ or misspecified responses $y_{\text{rejected}} \sim \pi_{\text{ref}}(y|x,c')$. We show that MIPO encourages models to adapt to individual user contexts rather than relying on general information in the prompt. This also serves as a useful test bed for non-verifiable tasks that RLVR cannot effectively handle, which is a key aspect of our motivation.

Our empirical results with 1-7B parameter Llama and Qwen instruct models show that MIPO improves personalization by 3–16% compared to personalized prompting baselines (Qwen2.5-1.5B-Instruct by 51%). We evaluate five different models on one pluralistic benchmark (Multi-Bench (Lee et al., 2024b)) and two real-user datasets(PRISM (Kirk et al., 2024a) and Community Alignment (Zhang et al., 2025c)).

Surprisingly, MIPO also shows gains in verifiable domains *without any verifiers or additional data*. We experiment on a suite of reasoning and MCQ tasks (Gu et al., 2025), including GSM (Cobbe et al., 2021) and the AI2 Reasoning Challenge (Clark et al., 2018), which the models are likely to be already finetuned on. Yet we observe that MIPO, with no

additional verifiable rewards, improves the performance of instruction-tuned models by 1–4% on average across tasks, and by as much as 20% for 1B Llama models. Furthermore, the performance gains achieved by MIPO often match or occasionally exceed RLVR which has access to privileged information.

In summary, our contributions include:

- Proposing MIPO, a novel self-training method based on contrastive data augmentation and DPO;
- Showing that the implicit reward in DPO is the pointwise mutual information between prompts and model outputs under the base LLM;
- Evaluating context-driven personalization on two real-user datasets and one personalized instruction-following benchmark, and showing that MIPO achieves 3–51% empirical gains over the prompting baselines;
- Applying MIPO to verifiable LLM benchmarks, such as GSM and the AI2 Reasoning Challenge, and showing an average improvement of 1–4% on top of instruction fine-tuned models and up to 20% for smaller models.

## 2. Related Work

Compared to prior work on data construction for DPO (Dwaracherla et al., 2024; Das et al., 2025; Gou & Nguyen, 2025; Qi et al., 2025; Yang et al., 2024; Zhu et al., 2025; Yao et al., 2024; Geng et al., 2025; Xu et al., 2024; D'Oosterlinck et al., 2024), we assume no external signals, stronger models, or ground-truth to guarantee the optimality of the chosen over the rejected responses. MIPO only requires that the chosen responses are generated from the correctly specified distribution, either by conditioning on the correct prompt or by conditioning on the correct prompt and context pair. Therefore, our work builds on self-training and self-improvement though many existing methods to this end also require verifiable rewards or human-supervised rubrics (Dong & Ma, 2025; Chen et al., 2024b; Singh et al.,

2024; Ulmer et al., 2024; Liang et al., 2024; Hosseini et al., 2024; Yuan et al., 2023). Yuan et al. (2025); Fränken et al. (2024) are exceptions, as they use the same model to reward the learned policy or generate training data. LLM personalization is also a growing field, but most existing methods rely on prompting or training with human preferences (Sorensen et al., 2024; Kirk et al., 2024a; Zhang et al., 2025c; Sorensen et al., 2025; Lee et al., 2024b; Kim et al., 2025; Zhao et al., 2025; Jiang et al., 2025a; Li et al., 2025; 2024; Liu et al., 2025; Poddar et al., 2024; Nam et al., 2026). In contrast, we do not require human supervision, and instead, focus on training models through intrinsic signals to effectively personalize responses based on the user-specific context given in a (context, prompt) pair. Due to space, we refer the reader to a more detailed discussion in Appendix A.

## 3. Preliminaries

**Post-training LLMs.** Let $\mathcal{D}$ be a dataset of annotated preferences consisting of a prompt $x$, and a corresponding pair of chosen and rejected responses to the same prompt $y_c$ (chosen) and $y_r$ (rejected). The Bradley Terry reward model $r$ assumes the following preference likelihood:

$$p(y_c \succ y_r | x) = \frac{\exp r(x, y_c)}{\exp r(x, y_c) + \exp r(x, y_r)}, \quad (1)$$

where the probability that $y_c$ is preferred over $y_r$ is a logistic function of the reward difference. The goal of preference optimization is to first learn this reward model $\hat{r}(x, y)$ using (human) labeled data, then learn a policy $\pi_\theta(y|x)$ that optimizes the learned reward subject to a KL-divergence constraint against the pre-trained reference policy $\pi_{\text{ref}}$. This leads to the following RLHF objective (Jaques et al., 2017; Ouyang et al., 2022):

$$\max_{\pi_\theta} \mathbb{E}_{x \sim \mathcal{D}, y \sim \pi_\theta(.|x)} \left[ \hat{r}(x, y) \right] - \beta \mathbb{D}_{KL} \left[ \pi_\theta || \pi_{\text{ref}} \right]. \quad (2)$$

For personalized instruction-following [1], we additionally define $c$ as the user-specific context, and modify the policy to condition on both the prompt $x$ and the context $c$, $\pi_\theta(y|x, c)$. Similarly, the reward is defined over both the prompt and the context as $r(x, y, c)$, consistent with prior work (Poddar et al., 2024; Li et al., 2024; Nam et al., 2026).

**Supervised Fine-tuning (SFT)** minimizes the forward KL divergence between the learned policy and the data generating policy (Xiao et al., 2025), and the loss is computed over the chosen responses:

$$\mathcal{L}_{\text{SFT}}(\theta; \mathcal{D}) = -\mathbb{E}_{(x, y_c) \sim \mathcal{D}} \left[ \log \pi_\theta(y_c|x) \right]. \quad (3)$$

---

[1]We call it personalized instruction-following to clarify that the model is given user-specific context $c$ along with query $x$, and the goal is to generate a response about $x$ that follows the user's preference in $c$.

**Direct Preference Optimization** (DPO) (Rafailov et al., 2024) is an alternative technique for learning from preference feedback that uses the same Bradley-Terry objective to directly modify the probabilities of the generator language model. The optimal policy $\pi_\theta$ is obtained from a reference model $\pi_{\text{ref}}$ via an energy re-weighting (Lv et al., 2025):

$$\pi_\theta(y|x) = \pi_{\text{ref}}(y|x) \frac{\exp\left(\frac{1}{\beta} r(x, y)\right)}{Z(x)}, \quad (4)$$

where $r$ is the reward, $\frac{1}{\beta} > 0$ is temperature, and $Z(x) = \mathbb{E}_{y \sim \pi_{\text{ref}}} \left[ \exp\left(\frac{1}{\beta} r(x, y,)\right) \right]$. This simplifies the two stage RLHF process of finding the reward, then an optimal policy to maximize this reward, into a single step:

$$\mathcal{L}_{\text{DPO}} = -\mathbb{E}_{\mathcal{D}} \left[ \log \sigma \left( \beta \log \frac{\pi_\theta(y_c|x)}{\pi_{\text{ref}}(y_c|x)} - \beta \log \frac{\pi_\theta(y_r|x)}{\pi_{\text{ref}}(y_r|x)} \right) \right] \quad (5)$$

**Contrastive Representation Learning** has been widely explored to learn latent representations that embed positive pairs close to each other and negative pairs far from each other. Most relevant to our work, van den Oord et al. (2019) proposes the Information Noise-Contrastive Estimation (InfoNCE) loss, which Poole et al. (2019) modifies as:

$$\mathcal{L}_{\text{infoNCE}} = -\mathbb{E}_{\mathcal{D}} \left[ \frac{1}{K} \sum_{i=1}^{K} \log \frac{\pi_\theta(y_i|x, c_i)}{\frac{1}{K} \sum_{j=1}^{K} \pi_\theta(y_i|x, c_j)} \right], \quad (6)$$

Minimizing this loss is equivalent to maximizing a lower bound on the mutual information between $X, Y$, where the positives are sampled from conditional distribution $p(y|x)$ and the negatives are sampled from marginal distribution $p(y)$. This objective has been adapted by numerous other works in RL and representation learning (Chen et al., 2020; Eysenbach et al., 2023; Mazoure et al., 2022; Hejna et al., 2024).

Fränken et al. (2024) extend InfoNCE to LLM settings in their method SAMI by maximizing the conditional mutual information between constitutions and responses, given by:

$$\mathcal{L}_{\text{SAMI}} = -\mathbb{E}_{\mathcal{D}} \left[ \frac{1}{2K} \sum_{i=1}^{K} \left( \log \frac{\pi_\theta(y_i|x, c_i)}{\frac{1}{K} \sum_{j=1}^{K} \pi_\theta(y_i|x, c_j)} \right. \right.$$
$$\left. \left. + \log \frac{\pi_\theta(y_i|x, c_i)}{\frac{1}{K} \sum_{l=1}^{K} \pi_\theta(y_l|x, c_i)} \right) \right] \quad (7)$$

While SAMI and MIPO are both motivated by InfoNCE, SAMI only applies to settings where prompts and constitutions are decoupled and therefore cannot be used in verifiable benchmarks. In contrast, MIPO can flexibly incorporate different data augmentation techniques (generating from prompts alone, or from prompts and contexts) and is built directly on top of DPO, making it an easy plug-in pre-processing step before standard DPO training.

# 4. Mutual Information Preference Optimization (MIPO)

We propose two instantiations of MIPO based on the same underlying principle: maximizing the mutual information between prompts and model responses as an intrinsic reward signal. First, we introduce a simplified version that maximizes the mutual information between model outputs and prompts *under the base policy* to make the connection with contrastive representation learning clear. Then we will modify this objective to maximize the conditional mutual information between responses and user contexts given the prompts.

**Setup:** For a given prompt $x \in \mathcal{X}$, we first sample a chosen response conditioned on the correct prompt: $y_c \sim \pi_{\text{ref}}(y|x)$, and pair it with a rejected response generated using a random prompt $y_r \sim \pi_{\text{ref}}(y|x'), x' \neq x$. We use this contrastive dataset of $(x, y_r, y_c)$ to train DPO.

**MIPO uses the pointwise mutual information under the base policy as an implicit reward.** DPO maximizes an implicit reward, replacing the two-step process used in traditional RLHF methods with single-step policy optimization. In typical RLHF settings, the training data comprises human-labeled preferences over chosen and rejected responses. However, in MIPO, *what learning signal do randomly paired rejected responses provide relative to the chosen responses?* The key lies in DPO's connection to InfoNCE (van den Oord et al., 2019). InfoNCE maximizes a lower bound on the mutual information $I(X, Y)$:

$$I(X;Y) \geq \log(N) + \underbrace{\mathbb{E}_X \left[ \log \frac{\exp r(x,y)}{\sum_{y_i \in \mathcal{Y}} \exp r(x,y_i)} \right]}_{-\mathcal{L}_{\text{infoNCE}}}, \quad (8)$$

where $\mathcal{Y}$ is a set of $N$ random samples containing one positive from the conditional distribution, $y_c \sim \pi_{\text{ref}}(y|x)$, and $N - 1$ negative samples from the marginal, $y_r \sim \pi_{\text{ref}}(y)$. Importantly, van den Oord et al. (2019) (Section 2.3 & A.1) show that the optimal critic is proportional to the density ratio of the conditional and the marginal distributions:

$$r(x,y) \propto \log \frac{\pi_{\text{ref}}(y|x)}{\pi_{\text{ref}}(y)}, \quad (9)$$

which is also the pointwise mutual information between $x, y$ under the reference model. In the case of one negative sample, InfoNCE becomes:

$$\mathcal{L}_{\text{infoNCE}} = -\mathbb{E}_{(x,y_r,y_c) \sim \mathcal{D}} \left[ \log \frac{\exp r(x,y_c)}{\exp r(x,y_r) + \exp r(x,y_c)} \right]. \quad (10)$$

Rewriting Eq. (4) in terms of the reward yields:

$$r(x,y) = \beta \log \frac{\pi_\theta(y|x)}{\pi_{\text{ref}}(y|x)} - \log Z(x), \quad (11)$$

and substituting this into the one-sample InfoNCE loss recovers the DPO objective. Therefore, when chosen responses are sampled according to $\pi_{\text{ref}}(y|x)$ and rejected responses are sampled from the marginal $\pi_{\text{ref}}(y)$[2], DPO with MIPO's contrastive pairs optimizes the following objective:

$$\arg\max_{\pi_\theta} \mathbb{E}_{x \sim \mathcal{X}, y \sim \pi_\theta(y|x)} \underbrace{\left[ \log \frac{\pi_{\text{ref}}(y|x)}{\pi_{\text{ref}}(y)} \right]}_{\text{optimal reward } r(x,y)} - D_{\text{KL}}(\pi_\theta || \pi_{\text{ref}}). \quad (12)$$

By optimizing this reward, we maximize the pointwise mutual information between prompts and model outputs under $\pi_{\text{ref}}$. This drives the learned policy $\pi_\theta$ to upweight responses likely under $\pi_{\text{ref}}(y|x)$ and downweight responses that are globally likely under the marginal $\pi_{\text{ref}}(y)$. Note that the reward is static and defined in terms of the base model $\pi_{\text{ref}}$, so the mutual information being maximized is with respect to the reference policy rather than the learned policy $\pi_\theta$. We hypothesize that this is actually a strength of MIPO. $\pi_{\text{ref}}$ provides a reliable model of natural language, so training the policy to maximize this fixed reward effectively steers $\pi_\theta$ to adapt to prompts. In contrast, using an on-policy version of mutual information may incentivize reward hacking by manipulating the learned policy's output distributions without affecting the underlying actions. To confirm this empirically, we conduct ablations with different approximations of mutual information under $\pi_\theta$ (PPO) and $\pi_{\text{ref}}$ (DPO).

## 4.1. Maximizing Conditional Mutual Information of Contexts and Responses Given Prompts

We use the mutual information between responses $y$ and user contexts $c$ conditioned on the prompts $x$ to learn an optimal user-conditioned policy $\pi_\theta(y|x,c)$. For example, given a user's query $x$ = *"Explain random variables"*, there could be two different user contexts. $c$ = *"I have a math PhD"*, and $c'$ = *"I am in 7th grade and just learned about probability"*. The goal of personalization is not only to respond to the prompt $x$, but also to address the user-specific information captured in $c$. This changes the mutual information in Eq. (8) to the following conditional objective:

$$I(Y;C|X) \geq \mathbb{E}_X \left[ \log \frac{\exp r(x,y_c,c)}{\sum_{y_i \in \mathcal{Y}} \exp r(x,y_c,c) + \exp r(x,y_r,c)} \right]. \quad (13)$$

This yields the following optimal critic:

$$r^*(x,y,c) \propto \log \frac{\pi_{\text{ref}}(y|x,c)}{\pi_{\text{ref}}(y|x)}. \quad (14)$$

---

[2]Since we cannot sample directly from the marginal, we use Monte Carlo approximation: sample random prompts $x'$, then generate $y_r \sim p(y|x')$. See Section 4.2 for a discussion of this approximation.

Maximizing the conditional pointwise mutual information $I_{\text{ref}}(y; c|x)$ improves the model's steerability by encouraging $\pi_\theta(y|x, c)$ to focus on user-specific context $c$ rather than relying on generic information in $x$ that works for any user.

### 4.2. Different Negative Sampling Approaches

While chosen responses are always generated with the correct prompt $x$ (or correct prompt and context $(c, x)$), we consider different strategies for sampling the negatives to pair with each positive response. We conduct ablations with (i) missing, (ii) random and (iii) reshuffling for personalized instruction-following, and (i) random and (ii) reshuffling for verifiable tasks that do not decouple prompts and contexts.

**When prompts and contexts are distinguished, negatives can be generated in the following two ways**:

$$y_r \sim \begin{cases} \pi_{\text{ref}}(y|x) & \text{(i) } \textit{missing} \text{ context} \\ \pi_{\text{ref}}(y|x, c'), c' \sim \mathcal{C} & \text{(ii) } \textit{random} \text{ context} \end{cases}$$

Strategy (i) samples directly from the prompt only, without the context. Strategy (ii) first samples a random user context $c'$, then generates a response conditioned on the prompt and the mismatched context $c'$. Since $\pi_{\text{ref}}(y|x) = \mathbb{E}_{c'}[\pi_{\text{ref}}(y|x, c')]$, Strategy (ii) is a unbiased estimator of $\pi_{\text{ref}}(y|x)$, but each negative sample $y_r$ is conditioned on $c'$. The second approach presents a trade-off. On the one hand, this may sharpen the contrast between chosen and rejected responses, since the rejected response is anchored to a different context. However, conditioning on the additional context introduces approximation error that vanishes only in expectation over many sampled contexts, so using a single negative may provide a coarse approximation.

**When prompts and contexts are not decoupled, negatives cannot be directly sampled from the marginal.** Since we do not have access to the marginal distribution $\pi_{\text{ref}}(y)$, we approximate sampling from the true marginal with $\mathbb{E}_{x' \sim \mathcal{X}}[\pi_{\text{ref}}(y|x)]$. We first sample a random prompt $x'$, then sample $y_r \sim \pi_{\text{ref}}(y|x')$. Pairing each positive sample with multiple negatives is more desirable than a single negative, since averaging across many $x'$s reduces the approximation error from conditioning on a particular $x'$. We conduct ablations with different numbers of negatives to test whether theory holds in practice, though we expect learning gains to plateau after some point.

We propose reshuffling the chosen responses as a practical alternative to re-generating with random prompts. While reshuffling violates the MIPO assumption that chosen and rejected responses are sampled independently from their respective distributions, it substantially reduces data-generation costs, which makes it appealing in practice. This also provides a clean comparison to SFT baselines, since SFT and MIPO (with reshuffling) use the exact same train-ing data. Both use the same chosen responses, but MIPO takes the additional step of reshuffling them to serve as rejected responses paired with the originals. This allows us to empirically isolate the contribution of the contrastive signal from the benefit of additional training data, since if the rejected responses were generated independently, MIPO would have $2\times$ the training data.

## 5. Experiments

### 5.1. Baselines

**Personalized Prompting.** For personalized instruction-following, models are prompted with additional user context $c$ along with prompt $x$ to personalize their responses. The default baselines are only conditioned on prompts $x$.

**Supervised Fine-Tuning (SFT)** is implemented in two ways: SFT without revision and SFT with revision. SFT without revision uses the same training data as MIPO (reshuffling). SFT with revision uses additional information to refine the model's initial answers. **Revision based on privileged information** (in verifiable tasks) uses ground-truth solutions: models are given the correct final answers and asked to revise their original answers accordingly. This privileged information is not available to MIPO or other unsupervised methods. **Revision based on contrastive answers** (in open-ended domains) improves the model's initial (chosen) response by comparing it to the rejected response and refining it (see the prompts for self-revision in Appendix D.2). The rejected response used for comparison is generated with the same prompt but a random context.

**RL with Verifiable Rewards (RLVR)** is only applicable in verifiable tasks. We use exact string matching on the extracted model response to reward the correct response by 1 and 0 otherwise. RLVR uses privileged information not given to MIPO, so we expect it to outperform other methods including MIPO. We are interested in how much of the gains can be matched by MIPO without any verifier signal.

**RL with AI Feedback** (RLAIF) (Bai et al., 2022a; Lee et al., 2024a) uses LLM-as-a-judge to assign rewards. The judge receives the user's query and context along with the model's response to give a score between 1–5 based on how well the response addresses the user-specific context. For verifiable tasks, we provide the judge with the ground-truth answer to score the model's reasoning and answer on a scale of 1–5. Rather than using a stronger critic model, we use the same, but frozen, model as the critic to enable a fair comparison, since no other method is trained with signals from a larger model. RLVR and RLAIF are both optimized with PPO.

## 5.2. Mutual Information Alternatives

For personalized instruction-following, where contexts and prompts are decoupled, we compare MIPO to other methods based on conditional mutual information. Specifically, we implement the following baselines:

**(i) MI-PPO** uses PPO to maximize the pointwise conditional mutual information under the learned policy as:

$$r_{\mathrm{MI}}(x, y, c) = \log \pi_\theta(y|x, c) - \log \pi_\theta(y|x). \quad (15)$$

**(ii) MI-RLAIF** combines mutual information reward with the task-specific reward from LLM-as-a-judge as follows:

$$r_{\mathrm{MI\_RLAIF}}(x, y, c) = r_{\mathrm{LLM\_as\_judge}}(x, y, c) + \alpha r_{\mathrm{MI}}. \quad (16)$$

**(iii) InfoNCE** (van den Oord et al., 2019), and **(iv) SAMI** (Fränken et al., 2024; Poole et al., 2019) are different instantiations of contrastive learning based on mutual information. We extend the original InfoNCE loss to LLM settings. SAMI (Eq. (7)) is closely related to InfoNCE but uses two symmetric loss terms: one contrasting different contexts for a fixed response, and the other contrasting different responses for a fixed context.

**Implementation.** Our experiments are implemented using OpenRLHF (Hu et al., 2024) and conducted with five different models (Qwen2.5-1.5/3/7B-Instruct (Qwen et al., 2025) and Llama-3.2-1/3B-Instruct (MetaAI, 2024)). Training details are in Appendix E and our code is available at `https://github.com/nam630/mutual_information_preference_optimization`.

## 5.3. Domains

We evaluate on 7 tasks including personalized instruction-following , math benchmarks (GSM8k (Cobbe et al., 2021) and SVAMP (Patel et al., 2021)), and multiple-choice question answering (MCQ) (AI2 Reasoning Challenge (ARC) (Clark et al., 2018)). For real-user datasets, we need to preprocess the data to obtain (context, query) pairs for training and evaluation. This helps simulate a setting where the user asks a general query along with specific requests or contexts, and the model needs to personalize the response. Details on the personalization tasks follow, and the LLM benchmarks are described further in Appendix D. We report final-answer accuracy on math and MCQs.

**Community alignment** (Zhang et al., 2025c) is a large-scale dataset collected from 3,000 participants across five countries, designed to capture pluralistic preferences and values. Since users select chosen and rejected responses without explicitly stating their ground-truth preferences, we use GPT-4o to infer the underlying preferences as user contexts for both training and testing. We sample 6,564 unique prompt-user pairs for training and 200 for evaluation.

**PRISM** (Kirk et al., 2024a) is a pluralistic dataset comprising open-ended conversations from 1,500 users across 75 countries and 20 LLMs. As above, we use GPT-4o to infer user contexts from ground-truth chosen and rejected responses. The models are tested on how well they address the user-specific context when given a (context, query) pair. We sample 7,294 data points for training and 194 for testing.

**Multifaceted Bench** (Lee et al., 2024b) is a benchmark for evaluating models' ability to generate context-specific responses. It includes 921 instructions from five existing benchmarks (AlpacaEval 2.0 (Dubois et al., 2025), FLASK (Ye et al., 2024), Koala (Arnav, 2023), MT-Bench (Zheng et al., 2023), and Self-Instruct (Wang et al., 2023)), each paired with synthetic, human-verified system messages specifying user preferences along four dimensions: style, background knowledge, informativeness, and harmlessness. We use GPT-4o to rewrite each system message into four first-person user messages, one per preference dimension, and split the dataset into 3,600 prompt-context pairs for training and 84 for testing.

## 5.4. Personalization Evaluation with LLM-judges 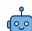

We use larger LLMs from the same model family (Qwen2.5-14B-Instruct for the Qwen models, and Llama-3-8B-Instruct for the Llama models) as judges to score responses (see LLM-judge prompts in Appendix D.2). To validate the judges' alignment with human preferences, we measure the tie-tolerant accuracy of predicting ground-truth human preferences from PRISM and Community Alignment: 8B Llama achieves 92% accuracy on both, and 14B Qwen achieves 93.5% and 92.2%, respectively. In our main results section, we report tie-inclusive win rates of the model-generated responses against ground-truth chosen responses (PRISM and Community Alignment datasets) and against personalized prompted GPT-4o responses (Multifaceted Bench).

**Validation with diverse LLM judges.** In additional experiments using a larger 70B model, we evaluate responses across a suite of stronger proprietary LLMs (GPT-4.1, GPT-5.2, Gemini 2.5 Flash, and Gemini 3.1 Flash Lite Preview). In the 70B experiment setup, each judge is shown two candidate responses (one from MIPO and the other from personalized prompting) and asked to select the preferred response according to the personalization rubric (see the LLM-judge prompt in Appendix D.2).

## 6. Results

### R1: MIPO improves personalized instruction-following by 3–51%, on average across datasets, over prompting.

Table 1 compares the win rates of MIPO-generated responses to other methods. Since the base models are already instruction-finetuned and receive information about

*Table 1.* 🤖 **Personalization win-rates evaluated on (i) Community Alignment (CA) (Zhang et al., 2025c), (ii) PRISM (Kirk et al., 2024a), (iii) Multi-Bench (MB) (Lee et al., 2024b).** Reported mean $\pm$ std from 3 seeds. Changes larger than 3 points from personalized prompting are in blue (for increase) and red (for decrease). We report the best number from different negative sampling strategies with MIPO.

| Model | CA | PRISM | MB |
|---|---|---|---|
| **Llama-3.2-1B-Instruct** | 31 | 36.6 | 59.52 |
| + Personalized Prompting | 78.00 | 72.17 | 79.76 |
| + SFT | $83.83 \pm 1.53$ | $69.76 \pm 1.07$ | $78.97 \pm 3.0$ |
| + SFT (Revision) | $80.00 \pm 1.00$ | $73.37 \pm 2.15$ | $77.78 \pm 3.00$ |
| + PPO (MI) | $80.17 \pm 3.33$ | $73.37 \pm 5.86$ | $82.22 \pm 4.05$ |
| + RLAIF | $79.83 \pm 5.69$ | $56.87 \pm 2.44$ | $84.12 \pm 0.69$ |
| + RLAIF (MI) | $81 \pm 2.50$ | $69.07 \pm 6.82$ | $82.93 \pm 1.82$ |
| + INFONCE | $89.5 \pm 2.00$ | $\mathbf{84.88 \pm 1.49}$ | $91.67 \pm 1.19$ |
| + SAMI | $91.17 \pm 0.76$ | $83.33 \pm 1.07$ | $92.86 \pm 1.19$ |
| + MIPO | $\mathbf{93.67 \pm 1.26}_{+15.7}$ | $80.93 \pm 3.61_{+8.8}$ | $\mathbf{93.26 \pm 0.69}_{+13.5}$ |
| **Llama-3.2-3B-Instruct** | 37 | 40.72 | 77.38 |
| + Personalized Prompting | 78 | 76.80 | 83.33 |
| + SFT | $84.17 \pm 1.61$ | $78.52 \pm 0.79$ | $90.48 \pm 1.19$ |
| + SFT (Revision) | $81.67 \pm 3.75$ | $78.35 \pm 1.86$ | $88.89 \pm 0.69$ |
| + PPO (MI) | $72.67 \pm 6.05$ | $73.02 \pm 1.66$ | $90.08 \pm 3.00$ |
| + RLAIF | $90.17 \pm 2.02$ | $80.07 \pm 2.84$ | $90.48 \pm 1.19$ |
| + RLAIF (MI) | $74.50 \pm 8.67$ | $80.58 \pm 1.19$ | $93.65 \pm 2.75$ |
| + INFONCE | $83.67 \pm 2.75$ | $78.69 \pm 0.60$ | $93.65 \pm 2.75$ |
| + SAMI | $\mathbf{91.5 \pm 2.29}$ | $82.99 \pm 0.1$ | $\mathbf{94.84 \pm 1.82}$ |
| + MIPO | $90.33 \pm 2.08_{+12.3}$ | $\mathbf{83.41 \pm 0.52}_{+4.7}$ | $\mathbf{94.84 \pm 1.82}_{+11.5}$ |
| **Qwen2.5-1.5B-Instruct** | 25.5 | 8.25 | 15.48 |
| + Personalized Prompting | 63.5 | 39.18 | 39.29 |
| + SFT | $60.5 \pm 0$ | $35.05 \pm 0.52$ | $39.68 \pm 3.00$ |
| + SFT (Revision) | $67 \pm 2.78$ | $46.56 \pm 0.79$ | $35.31 \pm 1.82$ |
| + PPO (MI) | $39.67 \pm 8.81$ | $32.47 \pm 3.61$ | $36.90 \pm 2.06$ |
| + RLAIF | $50.33 \pm 7.09$ | $13.92 \pm 3.09$ | $39.29 \pm 5.46$ |
| + RLAIF (MI) | $59.83 \pm 5.84$ | $16.83 \pm 4.79$ | $34.52 \pm 2.38$ |
| + INFONCE | $58.83 \pm 1.04$ | $50.69 \pm 2.93$ | $59.52 \pm 2.06$ |
| + SAMI | $72.83 \pm 1.26$ | $\mathbf{60.65 \pm 1.58}$ | $64.69 \pm 1.82$ |
| + MIPO | $\mathbf{78.83 \pm 2.84}_{+15.3}$ | $60.31 \pm 1.26_{+21.1}$ | $\mathbf{74.60 \pm 3.00}_{+35.31}$ |
| **Qwen2.5-3B-Instruct** | 34 | 11.86 | 46.43 |
| + Personalized Prompting | 76 | 49.49 | 63.10 |
| + SFT | $74.33 \pm 0.29$ | $44.67 \pm 0.79$ | $64.29 \pm 3.16$ |
| + SFT (Revision) | $74.83 \pm 0.29$ | $53.09 \pm 0.52$ | $64.68 \pm 2.48$ |
| + PPO (MI) | $72.67 \pm 6.05$ | $50.52 \pm 6.08$ | $66.67 \pm 1.19$ |
| + RLAIF | $77.33 \pm 2.08$ | $48.63 \pm 1.30$ | $70.64 \pm 2.75$ |
| + RLAIF (MI) | $74.50 \pm 8.67$ | $51.72 \pm 3.94$ | $72.22 \pm 2.48$ |
| + INFONCE | $70 \pm 1.73$ | $50.52 \pm 0.52$ | $73.02 \pm 3.00$ |
| + SAMI | $\mathbf{80.33 \pm 2.57}$ | $59.62 \pm 0.30$ | $71.83 \pm 4.51$ |
| + MIPO | $\mathbf{80.33 \pm 1.76}_{+4.3}$ | $\mathbf{60.31 \pm 1.36}_{+10.3}$ | $\mathbf{74.60 \pm 0.69}_{+18.2}$ |
| **Qwen2.5-7B-Instruct** | 39 | 20.62 | 47.62 |
| + Personalized Prompting | 78.5 | 58.25 | 70.24 |
| + SFT | 77 | 56.70 | 75 |
| + SFT (Revision) | 77 | 59.28 | 72.62 |
| + PPO (MI) | 79 | 54.12 | 72.62 |
| + RLAIF | 80.5 | $\mathbf{63.40}$ | 76.19 |
| + RLAIF (MI) | $\mathbf{86}$ | 64.43 | $\mathbf{78.57}$ |
| + INFONCE | 82.5 | 57.73 | 72.62 |
| + SAMI | 81.5 | 57.73 | 73.81 |
| + MIPO | $81.5_{+3}$ | $58.76_{+0.5}$ | $73.81_{+3.57}$ |

the user-specific context, personalized prompting offers a strong baseline relative to default prompting, which produces generic answers regardless of the user. As a result, personalized prompting (yellow) nearly doubles the win rate over the default responses. MIPO achieves substantial gains on top of personalized prompting: Qwen2.5-7B-Instruct improves by 3% on average across three domains, the 1–3B models improve by 13–16%, and Qwen2.5-1.5B-Instruct

in particular achieves a 51% improvement. While RLAIF provides a compelling baseline for some larger models, it performs poorly at the 1B scale. We hypothesize that this is due to unreliable reward signals from smaller critics (see Appendix 7).

Online mutual information approaches also fall short of MIPO. Both PPO (MI), which rewards mutual information only, and RLAIF (MI), which combines mutual information with the LLM-as-judge reward, underperform across most settings. This aligns with our hypothesis that maximizing the pointwise mutual information under the reference policy serves as a better empirical objective than maximizing the mutual information under the learned policy, since MIPO is less prone to reward hacking. MIPO also outperforms other InfoNCE-based MI implementations across most domains and models, though SAMI also performs comparably.

Examples in Appendix C show qualitative improvements in MIPO responses. For example, the prompt presents a conversation between two colleagues discussing lunch options: one speaker says they'll join their co-worker at a coffee shop to get a salad (*"No, I think I'll come with you. I'm longing for a nice salad."*). The model is asked to discuss the speaker's motivation, with specific instructions to consider factors such as social dynamics between colleagues and emotions driving the speaker's response. Personalized prompted responses emphasize the speaker's desire for a salad rather than social politeness or companionship. In contrast, MIPO suggests motivations grounded in the specific aspects the user asked about.

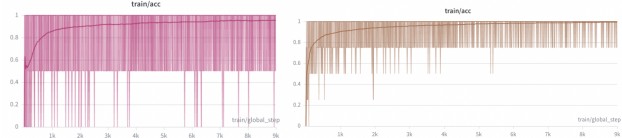

*Figure 2.* **Learning curves (positive and negative prediction accuracy) during training.** Left: Qwen2.5-3B-Instruct; right: Llama3-1B-Instruct on Multifaceted Bench with $N = 10$ negatives per positive. The learning curves eventually plateau around 6–7k training steps, suggesting that the gains of self-improvement eventually saturate.

**R2: MIPO improves models' output diversity via personalization.** Maximizing conditional MI increases diversity because variation in $c$ produces different outputs $y$ even for the same prompt. Self-BLEU-4 scores (Zhu et al., 2018) in Table 3 confirm this empirically. MIPO lowers average self-BLEU for all models. In contrast, SFT yields higher self-BLEU on average (see Table 4 for individual task-model pair scores). This result is especially meaningful given recent observations of LLM homogeneity (Jiang et al., 2025b; Zhang et al., 2025b; Kirk et al., 2024b; Abdulhai et al., 2026), since it suggests that personalization and in-context steerability could help mitigate mode collapse.

*Table 2.* **Reasoning benchmarks.** GSM and SVAMP use 8-shot; otherwise zero-shot. Reported mean and stddev from 3 seeds. Changes larger than 3 points are highlighted in red (+) and blue (-). Ablations with different negative sampling strategies are reported in Appendix 5.

| Model | GSM (8-shot) 📊 | SVAMP (8-shot) 📊 | ARC-Easy 🅰 | ARC-Challenge 🅰 | Avg. |
|---|---|---|---|---|---|
| **Llama-3.2-1B-Instruct** | 22 | 51.67 | 44 | 33.2 | 37.72 |
| + SFT | $24.83 \pm 1.61$+2.8 | $54.44 \pm 1.50$+2.8 | $34.13 \pm 5.5$-9.9 | $28.53 \pm 0.64$-4.7 | 35.48-2.2 |
| + MIPO (Ours) | $\mathbf{29.5 \pm 2.29}$+7.5 | $\mathbf{60.11 \pm 0.19}$+8.4 | $51.87 \pm 2.01$+7.9 | $39.53 \pm 3.11$+6.3 | 45.25+7.5 |
| + SFT (Ground-truth) | $24.17 \pm 1.04$+2.2 | $52.56 \pm 0.51$+0.9 | $28.53 \pm 1.10$-15.5 | $27.13 \pm 2.19$-6.1 | 33.10-4.6 |
| + RLAIF (Ground-truth) | $10.67 \pm 1.26$-11.3 | $52.56 \pm 2.50$+0.9 | $58.8 \pm 0.92$+14.8 | $42.2 \pm 1.04$+9.0 | 41.06+3.3 |
| + RLVR (Ground-truth) | $24.83 \pm 3.01$+2.8 | $55.89 \pm 3.10$+4.2 | $\mathbf{66.2 \pm 2.77}$+22.2 | $\mathbf{43.27 \pm 2.72}$+10.1 | $\mathbf{47.55}$+9.8 |
| **Llama-3.2-3B-Instruct** | 71.00 | 78.67 | 80.4 | 68.6 | 74.67 |
| + SFT | $64.5 \pm 1.32$-6.5 | $\mathbf{79.66 \pm 1.53}$+1.0 | $77.47 \pm 2.25$-2.9 | $63.67 \pm 0.50$-4.9 | 71.33-3.3 |
| + MIPO (Ours) | $70.17 \pm 2.02$-0.8 | $78.22 \pm 1.57$-0.4 | $\mathbf{85.26 \pm 0.58}$+5.8 | $\mathbf{70.93 \pm 0.7}$+2.3 | $\mathbf{76.15}$+1.5 |
| + SFT (Ground-truth) | $69.17 \pm 1.89$-1.8 | $77.21 \pm 1.00$-1.5 | $78.73 \pm 1.14$-1.7 | $70.53 \pm 1.45$+1.9 | 73.91-0.8 |
| + RLAIF (Ground-truth) | $\mathbf{71.67 \pm 1.76}$+0.7 | $79.11 \pm 2.99$+0.4 | $80.07 \pm 2.73$-0.3 | $70.33 \pm 1.80$+1.7 | 75.30+0.6 |
| + RLVR (Ground-truth) | $65.17 \pm 3.55$-5.8 | $79.44 \pm 0.51$+0.8 | $83.20 \pm 1.40$+2.8 | $69.60 \pm 1.20$+1 | 74.35-0.3 |
| **Qwen2.5-1.5B-Instruct** | 65.5 | 82.33 | 79 | 63.6 | 72.61 |
| + SFT | $67 \pm 3.50$+1.5 | $82.00 \pm 1.53$-0.3 | $66.07 \pm 1.51$-12.9 | $52.47 \pm 1.03$-11.1 | 66.89 |
| + MIPO (Ours) | $\mathbf{71 \pm 1.80}$+5.5 | $81.67 \pm 2.03$-0.7 | $82.27 \pm 0.81$+3.3 | $65.93 \pm 1.70$+2.3 | 75.22+2.4 |
| + SFT (Ground-truth) | $67.50 \pm 2.00$+2.0 | $81.33 \pm 0.34$-1.0 | $75.2 \pm 0.87$-3.8 | $62.2 \pm 1.83$-1.4 | 71.56-1.3 |
| + RLAIF (Ground-truth) | $62.5 \pm 7.76$-3.0 | $\mathbf{83 \pm 1.67}$+0.7 | $85.4 \pm 0.2$+6.4 | $\mathbf{70.47 \pm 0.76}$+6.9 | $\mathbf{75.34}$+2.5 |
| + RLVR (Ground-truth) | $62.33 \pm 5.97$-3.2 | $79.78 \pm 0.96$-2.6 | $\mathbf{86.73 \pm 0.70}$+4.4 | $69.13 \pm 3.59$+5.5 | 74.49+1.6 |
| **Qwen2.5-3B-Instruct** | 84.5 | 90.67 | 92 | 79.4 | 86.64 |
| + SFT | $85.83 \pm 0.76$+1.3 | $89.78 \pm 1.02$-0.9 | $87.47 \pm 0.23$-4.5 | $78.67 \pm 1.17$-0.7 | 85.44-1.2 |
| + MIPO (Ours) | $\mathbf{89.17 \pm 3.75}$+4.7 | $\mathbf{91.33 \pm 1.73}$+0.7 | $90.80 \pm 0.35$-1.2 | $\mathbf{80.13 \pm 2.01}$+0.7 | $\mathbf{87.86}$+1.2 |
| + SFT (Ground-truth) | $83.83 \pm 2.47$-0.7 | $90.56 \pm 0.20$-0.1 | $90.33 \pm 0.99$-1.7 | $79.6 \pm 1.11$+0.2 | 86.08-0.5 |
| + RLAIF (Ground-truth) | $82.83 \pm 1.53$-1.7 | $90.22 \pm 0.69$-0.5 | $91.8 \pm 0.2$-0.2 | $79.4 \pm 1.25$ | 86.06-0.6 |
| + RLVR (Ground-truth) | $69.83 \pm 4.31$-14.7 | $85.89 \pm 4.40$-4.8 | $91.47 \pm 0.46$-0.5 | $78.60 \pm 0.60$-0.8 | 81.45-5.2 |
| **Qwen2.5-7B-Instruct** | 93.5 | 91.33 | 93.80 | 88.20 | 91.71 |
| + SFT | $\mathbf{92.50}$-1.0 | $92.00$+0.7 | $93.60$-0.2 | $87.80$-0.4 | 91.48-0.2 |
| + MIPO (Ours) | $\mathbf{93.00}$-0.5 | $91.67$+0.3 | $93.80$ | $\mathbf{90.4}$+2.2 | $\mathbf{92.22}$+0.5 |
| + SFT (Ground-truth) | $92.00$-1.5 | $91.33$ | $93.60$-0.2 | $90.00$+1.8 | 91.73 |
| + RLAIF (Ground-truth) | $92.50$-1.0 | $91.33$ | $\mathbf{94.20}$+0.4 | $88.60$+0.4 | 91.66-0.1 |
| + RLVR (Ground-truth) | $90.5$-3 | $90.67$-0.7 | $\mathbf{94.20}$+0.4 | $86.4$-1.8 | 90.44-1.3 |

*Table 3.* **Self-BLEU-4 (Zhu et al., 2018) from pre- and post-training averaged across three personalization tasks.** Lower values indicate greater diversity (✓ indicates diversity improvement). See Table 4 for individual task-model scores.

| Model | Pre-Training | Post-SFT | Post-MIPO |
|---|---|---|---|
| Llama 1B | 0.420 | 0.427 ✗ | **0.393** ✓ |
| Llama 3B | 0.379 | 0.389 ✗ | **0.371** ✓ |
| Qwen 1.5B | 0.310 | 0.316 ✗ | **0.256** ✓ |
| Qwen 3B | 0.312 | 0.316 ✗ | **0.272** ✓ |
| Qwen 7B | 0.310 | 0.318 ✗ | **0.309** ✓ |

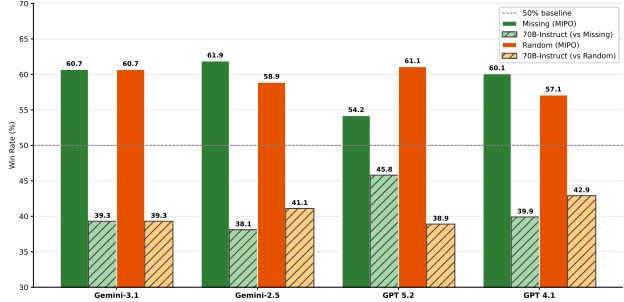

*Figure 3.* **Win rates of MIPO versus instruct-tuned models.** We report the win rates of MIPO against the baseline (y-axis) across a suite of LLM-judges (x-axis).

**R3: MIPO gains scale to 70B models, achieving 54–62% win rates against personalized prompting baselines.** We train Llama-3.3-70B-Instruct with two versions of MIPO (using missing and random contexts as negatives) on Multifaceted Bench using LoRA rank 32 (Hu et al., 2021), and compare both versions against the personalized prompting baseline. For evaluation, we present each pair of responses to four LLM judges (GPT-4.1, GPT-5.2, Gemini 2.5 Flash, and Gemini 3.1 Flash Lite Preview). Each judge selects the preferred response according to a personalization rubric based on the presented user context (see Appendix D.2 for the prompts). Fig. 3 shows that MIPO's benefits are consistently observed at the 70B scale across all four LLM judges.

So far, our empirical results have established MIPO as an effective personalization method that yields improvements without any human supervision or additional data. However, does MIPO generalize to tasks beyond personalization? Unlike prior work on constitution following (Fränken et al., 2024), MIPO extends to more general settings where prompts and contexts are not decoupled. When only prompts are given, chosen responses are paired with negatives generated from *random* prompts $x' \neq x$. We evaluate this approach on a range of LLM benchmarks, including math and MCQ. While we do not expect MIPO to outperform RLVR, since RLVR requires access to ground-truth

solutions and verifiers, we test whether MIPO can serve as a meaningful alternative despite having no external feedback.

**R4: Surprisingly, MIPO generalizes to math and MCQ tasks, achieving 1–4% improvements overall and an 20% improvement for smaller 1B Llama models.** In Table 2, the benefit of the contrastive signal is highlighted by the gap between SFT (without ground-truth) and MIPO. SFT performs poorly, likely because the smaller base models' initial outputs are of low quality. By contrast, MIPO's contrastive pairs yield substantial improvements over the instruction-following baselines, even at small model scales. MIPO also occasionally outperforms RLVR and RLAIF, both of which have access to ground-truth answers during training. MIPO shows impressive gains especially for smaller 1B models with raw point increases of 2.4–7.5 on average across tasks. Although performance gains vary by domain and model, we hypothesize that this reflects how heavily a particular model has been fine-tuned for specific tasks. For example, we suspect Qwen2.5-7B-Instruct has been heavily fine-tuned on math benchmarks like GSM, leaving little room for further improvement.

**R5: We investigate the effects of different negative sampling strategies** (Appendix Table 5). For personalized instruction-following, sampling rejected responses with random or missing contexts substantially outperforms reshuffling: random sampling outperforms reshuffling by 1.3–13% (varying across model and domain), and missing-context sampling outperforms it by 3–30%. This is likely because the conditional MI objective requires the chosen and rejected responses to differ only in user context, not in the prompt. Reshuffling violates this assumption by pairing responses with different prompts as well as different contexts. For math and MCQ benchmarks, by contrast, reshuffling performs comparably to random sampling. Although re-using chosen responses violates the independence assumption between positives and negatives, this does not appear to hurt empirical performance on these benchmarks.

To test whether negatives need to come from the reference policy, we conduct ablations by pairing chosen responses with random alphanumeric strings of the same length as the rejected responses (Appendix Table 6). Although these negatives still provide a clear contrastive signal against the positives, they fail to match MIPO's gains. Table 6 also shows that increasing the number of negatives per positive further improves MIPO's performance, which is consistent with the theoretical understanding that larger $N$ tightens the lower bound on mutual information.

## 7. Conclusion & Discussion

We propose MIPO, a contrastive data augmentation method based on maximizing mutual information between prompts

and responses, and evaluate it across a suite of both verifiable and non-verifiable tasks. Our empirical evaluation shows that:

- MIPO provides effective personalization, improving models by 3–51% across three pluralistic user datasets;
- MIPO also extends to a broader set of LLM tasks, including math and MCQ, achieving an additional 1–4% improvement on top of already instruction-finetuned models;
- MIPO yields especially large gains for smaller models whose self-generated data is likely suboptimal (e.g., 20% average improvement for Llama3.2-1B-Instruct).

Crucially, MIPO can achieve these gains *without any additional data or human supervision* and requires only a simple data-processing step before standard DPO training. These results suggest a promising direction for self-improvement using LLMs' implicit reward signals.

**Limitations. (i) Reliability of LLM judges.** Our current personalization results rely on LLM-as-judge for evaluation, which has both advantages and known limitations. While LLM-based evaluation is often used for cost-effectiveness, replicability, and high accuracy [3], it can also exhibit biases, such as the model's preference for verbosity. Therefore, when using LLM judges, it is crucial to validate them against human preference judgments or ground-truth data. **(ii) Using GPT-4o to obtain user contexts from real-world data.** For personalization, we used GPT-4o as a preprocessing step to obtain user contexts, which are not directly available in many real-user datasets (e.g., PRISM and Community Alignment). This information is made available to all baseline methods, and we evaluate only the policy's ability to adapt to these user-specific details in its response. An alternative to using GPT-4o would be to use past conversations or self-stated preferences. However, self-stated preferences are typically unavailable in real-world datasets unless explicitly collected via surveys (Kirk et al., 2024a). Moreover, even state-of-the-art reward models conditioned on self-stated preferences or past conversations achieve only 60-62% accuracy in preference prediction (Nam et al., 2026), which limits their reliability for judging personalized response preferability. We consider this a crucial limitation of many existing personalization datasets and see this as an important area of future work. **(iii) Noisy training signals.** Since MIPO's training signal is derived from the model itself, its effectiveness depends on the model's initial capabilities and the diversity of training prompts. When prompts or contexts are too similar to each other, the contrastive signal may be too weak to drive learning effectively.

---

[3]For example, AlpacaEval 2.0 (Dubois et al., 2025) uses GPT-4 as a judge and demonstrates strong correlation with human evaluations from Chatbot Arena (Spearman correlation of 0.93–0.98 based on 20K annotations).

## Impact Statement

Homogenization of model outputs is increasingly becoming an important AI safety concern (Abdulhai et al., 2026). We believe that more effective steering of model responses and improved personalization can help mitigate this issue. We propose a self-training method that maximizes the mutual information between prompts and model responses, and, in the personalization setting, the conditional mutual information between user contexts and model responses conditioned on the prompt. Our empirical results are promising in showing that models can self-improve without external feedback or additional data. However, this does not downplay the role of human oversight in model development. Human (preference) data remains critical but is often expensive to collect. We show that an implicit reward signal based on mutual information can serve as a useful alternative when such data is difficult to gather or scale. This may allow human data collection efforts to focus instead on model evaluation and deployment oversight, where they are most needed to ensure safe and aligned model behavior, and reduce the burden of human data in model training cycles.

## Acknowledgment

This research was supported by the UW-Amazon Science Gift Hub, UW-Tsukuba Amazon NVIDIA Cross Pacific AI Initiative (XPAI), Sony Research Award, Tinker Research Grants, Character.AI, DoorDash, Open Philanthropy, Coefficient Giving, Toyota Research Institute, and the Schmidt AI2050 Fellows program. This material is based upon work supported by the Defense Advanced Research Projects Agency and the Air Force Research Laboratory, contract number(s): FA8650-23-C-7316. Any opinions, findings and conclusions, or recommendations expressed in this material are those of the author(s) and do not necessarily reflect the views of AFRL or DARPA.

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

# A. Appendix: Related Work

**Reinforcement Learning (RL) fine-tuning of LLMs.** Post-training with RL has proven effective for tasks that target a model's *edge of competence*, i.e., tasks that are difficult but within the pre-training data distribution (Zhang et al., 2025a). Two popular frameworks are: RL with Verifiable Rewards (RLVR) (Lambert et al., 2025; Guo et al., 2025) and RL with Human Feedback (RLHF) (Jaques et al., 2019; Stiennon et al., 2022; Ouyang et al., 2022) based on whether the reward is verifiable or estimated from human preference data. Optimization can be implemented with different RL algorithms, including Proximal Policy Optimization (PPO) (Schulman et al., 2017), Group Relative Policy Optimization (GRPO) (Shao et al., 2024), and Direct Preference Optimization (DPO) (Rafailov et al., 2024).

**Self-training and improvement.** RL from AI feedback (RLAIF) reduces the burden of human preference data collection by using an LLM-as-a-judge (Bai et al., 2022b; Lee et al., 2024a) or self-rewarding models (Yuan et al., 2025). Other methods use multi-agent reinforcement learning (MARL) to train both the generator and the evaluator, allowing the generator to learn from the evaluator's reward signal (Dong & Ma, 2025; Chen et al., 2024b). However, these approaches still rely on verifiable rewards or human-annotated data in order to train a reliable evaluator. Similarly, self-supervised training methods often require external verifiers to ensure correctness of the training data or use human-generated rubrics for quality assessment (Singh et al., 2024; Ulmer et al., 2024; Liang et al., 2024; Hosseini et al., 2024; Yuan et al., 2023). While iterative refinement and self-feedback has shown success (Madaan et al., 2023; Shridhar et al., 2023; Zelikman et al., 2022), Huang et al. (2024) observe that self-correction can degrade model performance when external feedback or verifiers are not available. Tyen et al. (2024) speculate that this may be due to the model's limited capability to identify mistakes rather than to correct them. We also observe in our experiments that prompting models to self-revise and training on these revisions can cause performance degradation in some tasks.

**Direct Preference Optimization (DPO) with data augmentation.** Prior works have explored different data generation or selection methods to improve RLHF outcomes (Dwaracherla et al., 2024; Das et al., 2025; Gou & Nguyen, 2025; Qi et al., 2025; Yang et al., 2024; Zhu et al., 2025; Yao et al., 2024; Geng et al., 2025). Specifically for DPO, we highlight three data generation strategies from prior work: (1) Pairing suboptimal data for contrastive signals. Yao et al. (2024) empirically show that DPO works with *less-wrong-over-wrong* pairs, even when the chosen responses are also incorrect. Similarly, Xu et al. (2024); Geng et al. (2025) show that contrasting outputs from a strong model against those from a weaker model leads to model improvement. (2) Improving data through targeted revision. D'Oosterlinck et al. (2024) propose improving rejected responses along specific dimensions to obtain targeted, higher-quality chosen responses. (3) Bootstrapping from original data by applying perturbations in either positive or negative directions. When ground-truth responses are available, Samokhin et al. (2025) suggest pairing correct responses with random examples from the dataset to create negative pairs. Anderson et al. (2018) create negative samples by adding semantic perturbations to the original text. Yin et al. (2024) expand preference pairs based on semantic similarity of prompts to pair the chosen and rejected responses with additional prompts. While these works provide promising results suggesting benefits of textual data augmentation, they depend on larger models (Xu et al., 2024; Geng et al., 2025), human instructions for revision (D'Oosterlinck et al., 2024), or ground-truth data or verifiers to ensure the relative quality of the chosen over the rejected (Samokhin et al., 2025; Yin et al., 2024). In contrast, we show that models can self-improve without any external supervision or verifiers by contrasting responses conditioned on correct versus incorrect prompts.

**Personalization of LLMs.** Despite the success of making LLMs more helpful and performant (Bai et al., 2022a; Ouyang et al., 2022), recent work shows that human preferences are diverse and sometimes even conflicting (Sorensen et al., 2024; Kirk et al., 2024a; Zhang et al., 2025c). Therefore, preference alignment for LLMs should consider pluralistic or personalized alignment to individual preferences and traits. This capability is also known as *in-context adaption*, as models need to steer their outputs based on in-situ information gathered about users during conversations (Sorensen et al., 2025). To address this growing interest, many personalization benchmarks, using a mix of synthetic and real-user data, have been constructed (Lee et al., 2024b; Kim et al., 2025; Zhao et al., 2025; Jiang et al., 2025a; Li et al., 2025). A number of methods have also been developed to improve personalization, sharing the core idea of learning a user-conditioned policy or preference model that can distinguish among heterogeneous preferences (Li et al., 2024; Liu et al., 2025; Poddar et al., 2024; Nam et al., 2026)). Unlike prior approaches that rely on expert demonstrations or human preference datasets, we show that personalization can be achieved without human supervision by leveraging the model's intrinsic signals derived from contrastive response pairs. Closest to our work, Fränken et al. (2024) use an InfoNCE-derived loss for constitution following; we adapt their method to personalization as one of our baselines. Rather than introducing a new loss function, MIPO is built directly on DPO, a widely used post-training algorithm. As a result, MIPO has the practical advantage of reducing to a simple, plug-in data augmentation step before standard DPO training. Moreover, MIPO can be applied flexibly

to both personalization and general LLM tasks, showing empirical gains in both, unlike the prior approach which is limited to settings with decoupled prompts and constitutions.

# B. Appendix: Results

*Table 4.* **Self-BLEU-4 (Zhu et al., 2018) from pre- and post-training in (i) CA Community Alignment (CA), (ii) PRISM, (iii) Multi-bench (MB).** We report the mean from 3 seeds for all trained models, except Qwen-7B. Lower values mean higher diversity (✓ indicates diversity improvement).

| Model | CA | PRISM | MB | Avg. |
|---|---|---|---|---|
| **Llama-3.2-1B-Instruct** (Personalized-Prompting) | 0.393 | **0.356** | 0.510 | 0.420 |
| + SFT | **0.362** | 0.384 | 0.535 | 0.427 |
| + **MIPO-Personalized** | 0.363 | 0.365 | **0.450** | **0.393** ✓ |
| **Llama-3.2-3B-Instruct** (Personalized-Prompting) | **0.311** | **0.274** | 0.551 | 0.379 |
| + SFT | 0.330 | 0.297 | 0.539 | 0.389 |
| + **MIPO-Personalized** | 0.329 | 0.288 | **0.495** | **0.371** ✓ |
| **Qwen2.5-1.5B-Instruct** (Personalized-Prompting) | 0.208 | 0.167 | 0.554 | 0.310 |
| + SFT | 0.231 | 0.178 | 0.538 | 0.316 |
| + **MIPO-Personalized** | **0.195** | **0.150** | **0.423** | **0.256** ✓ |
| **Qwen2.5-3B-Instruct** (Personalized-Prompting) | 0.200 | 0.156 | 0.581 | 0.312 |
| + SFT | 0.225 | 0.158 | 0.564 | 0.316 |
| + **MIPO-Personalized** | **0.188** | **0.144** | **0.483** | **0.272** ✓ |
| **Qwen2.5-7B-Instruct** (Personalized-Prompting) | 0.197 | 0.158 | 0.575 | 0.310 |
| + SFT | 0.211 | **0.160** | 0.583 | 0.318 |
| + **MIPO-Personalized** | **0.193** | 0.163 | **0.570** | **0.309** ✓ |

*Table 5.* **Comparison of different negative sampling strategies for MIPO.** We consider three different ways of data augmentation: **(1) reshuffling**, **(2) random contexts or prompts**, and **(3) missing contexts**. For personalization, approach (2) pairs rejected $y_r \sim p(y|x, c')$ with chosen $y_c \sim p(y|x, c)$ for the query-context pair $(x, c)$; and approach (3) pairs rejected $y_r \sim p(y|x)$ with chosen $y_c \sim p(y|x, c)$.

| Model | CA | PRISM | MB | GSM | SVAMP | ARC Easy | ARC Challenge |
|---|---|---|---|---|---|---|---|
| **Llama-3.2-1B-Instruct** | 78 | 72.17 | 79.76 | 22 | 51.67 | 44 | 33.2 |
| + Reshuffling | 88.33 (2.02) | 77.84 (3.09) | 84.52 (2.38) | **29.50 (2.29)** | **60.11 (0.19)** | 51.87 (2.01) | **39.53 (3.11)** |
| + Random | 87 (5.57) | 77.32 (1.36) | 89.68 (1.92) | 27.33 (1.26) | 52.22 (1.02) | **53.47 (2.80)** | 27.33 (1.26) |
| + Missing | **93.67 (1.26)** | **80.93 (3.61)** | **93.26 (0.69)** | - | - | - | - |
| **Llama-3.2-3B-Instruct** | 78 | 76.80 | 83.33 | 71 | 78.67 | 80.4 | 68.6 |
| + Reshuffling | 86.63 (1.89) | 80.76 (2.84) | 91.27 (1.82) | 70.17 (2.02) | **78.22 (1.57)** | 85.26 (0.58) | 70.93 (0.70) |
| + Random | 88 (2.29) | **83.41 (0.52)** | **94.84 (1.82)** | 72.17 (0.29) | 76.89 (0.77) | **85.40 (0.69)** | **72.67 (1.81)** |
| + Missing | **90.33 (2.08)** | 81.62 (2.08) | 94.45 (2.48) | - | - | - | - |
| **Qwen2.5-1.5B-Instruct** | 63.5 | 39.18 | 39.29 | 65.5 | 82.33 | 79 | 63.6 |
| + Reshuffling | 71.83 (1.53) | 50.86 (0.78) | 40.87 (3.44) | **71 (1.80)** | **81.67 (2.03)** | **82.27 (0.81)** | 65.93 (1.70) |
| + Random | 68 (1.00) | **60.31 (1.36)** | 55.95 (3.57) | 70.33 (1.15) | 81.33 (1.20) | 81.87 (0.31) | **66.53 (1.10)** |
| + Missing | **78.83 (2.84)** | 59.79 (1.36) | **74.6 (3.0)** | - | - | - | - |
| **Qwen2.5-3B-Instruct** | 76 | 49.49 | 63.10 | 84.5 | 90.67 | 92 | 79.4 |
| + Reshuffling | 76.17 (1.76) | 56.19 (0.52) | 65.87 (4.51) | **89.17 (3.75)** | **91.33 (1.73)** | 90.80 (0.35) | 80.13 (2.01) |
| + Random | **80.33 (1.76)** | **60.31 (1.36)** | 70.63 (2.48) | 86.67 (2.25) | **91.33 (0.33)** | **91.13 (0.42)** | **80.80 (0.80)** |
| + Missing | 79.83 (0.29) | 59.79 (1.36) | **74.60 (0.69)** | - | - | - | - |

*Table 6.* **Increasing the number of negatives improves MIPO performance.** We trained MIPO with $N = 1, 3, 5, 10$ negatives, where rejected samples are generated using random contexts on Multifaceted Bench. We find that performance generally improves with larger $N$, as the MI lower bound tightens with increasing $N$. We also added random negatives which are random alphanumeric strings of the same lengths as the chosen responses. While the other negatives are drawn from the reference policy's distribution, these random negatives lie outside the policy's support and thus break the MIPO's theoretical assumption, despite providing clear contrastive signals with the chosen responses. Therefore, as expected, MIPO with random negatives performs worse than MIPO with theoretically aligned negatives. Best performance from each row is boldfaced.

| $N$ | Random | 1 | 3 | 5 | 10 |
|---|---|---|---|---|---|
| **Llama-3.2-1B-Instruct** | 79.8 | 89.7 | 89.3 | 91.7 | **94.1** |
| **Llama-3.2-3B-Instruct** | 90.5 | 94.8 | **95.2** | **95.2** | 92.2 |
| **Qwen2.5-1.5B-Instruct** | 34.5 | 56.0 | 54.8 | 57.1 | **58.3** |
| **Qwen2.5-3B-Instruct** | 64.3 | 70.6 | 75.0 | **81.0** | 78.6 |

*Table 7.* **Critic performance evaluated using ground-truth human preference labels.** We compute the strict and tie-tolerant accuracy of predicting a higher score for the chosen response over the rejected response.

| Critic model | Community Alignment (CA) | | PRISM | |
|---|---|---|---|---|
| | Strict acc. | Tie-tolerant acc. | Strict acc. | Tie-tolerant acc. |
| Llama-3.2-1B-Instruct | **6** | 98.5 | **36.5** | 76.5 |
| Llama-3.2-3B-Instruct | 54 | 89.5 | 72.5 | 87.5 |
| Llama-3-8B-Instruct | 76 | 92 | 68.5 | 92 |
| Qwen2.5-1.5B-Instruct | **27.5** | 87 | **42.5** | 78 |
| Qwen2.5-3B-Instruct | 6 | 100 | 74.5 | 91.5 |
| Qwen2.5-7B-Instruct | 63.5 | 98.5 | 72.46 | 92.6 |
| Qwen2.5-14B-Instruct | 63 | 93.5 | 78 | 91.5 |

While prior work on RLHF has also observed similarly moderate reward model accuracy of 69.6% and 72.4%, the learned policy still achieved substantial performance gains (see Appendix E.2 (Ouyang et al., 2022)). However, we suspect that an accuracy below 50% or unreliable rewards (i.e., the critic always gives 1 or 5 indiscriminately regardless of the response quality) is detrimental to learning and can lead to performance degradation after RLAIF.

In particular, we observe that Llama-1B on Community Alignment mostly assigns a score of 1 to both chosen and rejected responses; while Qwen-3B mostly assigns a score of 5. In both cases, the critic fails to discriminate between the chosen and rejected responses.

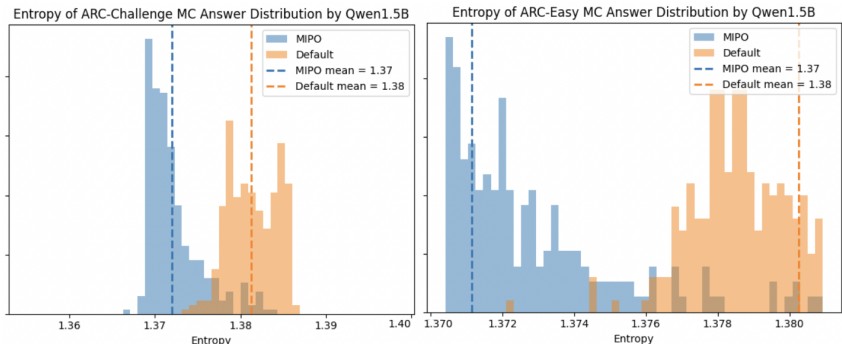

*Figure 4.* **Entropy over the MCQ answer choices conditioned on correct model predictions.** In addition to improving accuracy, MIPO (blue) also makes models become more confident about correct predictions compared to the base model (orange) as indicated by the mean and overall distribution shift. The x-axis ranges from mean $\pm$ 1 std.

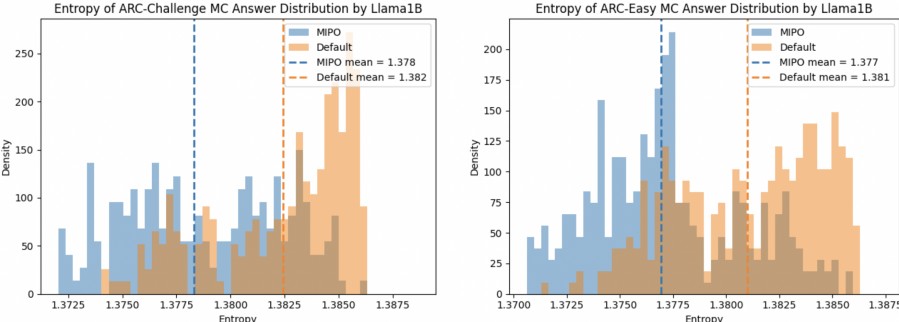

*Figure 5.* **Entropy over the 500 MCQ answer choices in ARC conditioned on output correctness.** MIPO-trained models (blue) become more confident about correct answers compared to the base models (orange) as indicated by entropy reduction. The range is truncated to show the mean $\pm$ 2 standard deviation.

# C. Appendix: MIPO-Personalized Responses

Examples from the Multi-Bench personalization benchmark

**Prompt:** I like answers that show sensitivity and consideration for different people's needs or preferences, especially when discussing things like food. That means being ***thoughtful about dietary restrictions, cultural tastes, or health-related matters.*** Given the following dialogue, answer to the question. [*Dialogue about people deciding on lunch.*], "Target": "No, I think I'll come with you. I'm longing for a nice salad.", "Question": "What is or could be the cause of target?"

**Default #1 (Llama3.2-3B-Instruct)**: In this dialogue, the cause of "target" is likely the coffee shop where they plan to go for lunch. The speaker, Louise, suggests going to the coffee shop, and the other person agrees to join her, indicating that the coffee shop is the target of their lunch plans.

**MIPO-Personalized #1 (Llama3.2-3B-Instruct)**: The cause of B's statement could be a few factors:

**1. **Dietary Preferences**:** B might prefer salads over other types of food, especially if they are health-conscious or follow a specific diet.

**2. **Cultural Tastes**:** B might have a cultural preference for salads, which aligns with their desire to eat one.

**3. **Health-Related Matters**:** B might be trying to maintain a healthy lifestyle and see the salad as a better option than a cheeseburger and fries.

**4. **Social Interaction**: B might want to spend more time with A and enjoy a meal together, and a salad might be a suitable choice that both can agree on.**

These factors collectively contribute to B's decision to join A for lunch and choose a salad.

**Prompt:** I like it when stories focus on the personalities and emotions of the characters. It helps me connect with them and understand their decisions better. Given the following dialogue, answer to the question. [*Dialogue about people deciding on lunch.*], "Target": "No, I think I'll come with you. I'm longing for a nice salad.", "Question": "What is or could be the cause of target?"

**Default #2 (Llama3.2-3B-Instruct):** Based on the dialogue, the cause of "I'm longing for a nice salad" could be that the person (B) is not getting enough variety in their diet or is craving something healthy, or possibly they are trying to maintain a healthy lifestyle.

**MIPO-Personalized #2 (Llama3.2-3B-Instruct):** Based on the dialogue, it seems that the cause of B's decision to accompany A to the coffee shop and order a salad could be a combination of several factors. Here are a few possibilities:

**1. **B's desire for a break from their usual routine**:** By saying "I usually bring my lunch from home and eat at my desk," B implies that they often spend most of their day working alone, which might be isolating. The invitation to go out and get lunch together could be a way for B to break up the monotony and socialize with A.

**2. **B's emotional state**:** The phrase "I'm longing for a nice salad" suggests that B might be feeling a bit down or stressed, and the thought of eating a salad at a coffee shop is a way for them to treat themselves and lift their mood.

**3. **B's desire to connect with A**:** By agreeing to go out with A, B might be seeking a chance to bond with their friend and have some quality time together. This could be a way for B to show that they value their relationship with A and want to spend time together.

**4. **B's curiosity about A's preferences**:** B's question about where A would like to eat and their subsequent decision to join A might also be driven by a desire to understand A's tastes and preferences. By trying the coffee shop with A, B can get a sense of what A enjoys and might even discover new things about their friend.

Overall, B's decision to accompany A to the coffee shop seems to be motivated by a mix of practicality, emotional needs, and a desire to connect with their friend.

---

Examples from the Multi-Bench personalization benchmark

**Prompt #3:** I'm pretty familiar with casual dining and how people tend to interact in social eating situations, so you don't have to explain the basics to me. Instead, include things that are more specific or relatable to the vibe of going out with friends or coworkers. Given the following dialogue, answer to the question. [*Dialogue about people deciding on lunch.*], "Target": "No, I think I'll come with you. I'm longing for a nice salad.", "Question": "What is or could be the cause of target?"

**Default #3 (Llama3.2-3B-Instruct):** The cause of "Target" in this dialogue is that Louise is suggesting that B should join her at the coffee shop, implying that the coffee shop is a suitable place for a casual lunch outing.

**MIPO-Personalized #3 (Llama3.2-3B-Instruct):** In this dialogue, the phrase "I'm longing for a nice salad" is a idiomatic expression that suggests B is craving a salad, but it's not just a simple desire for a healthy meal. The word "longing" implies a stronger emotional attachment or nostalgia for a salad, possibly because it's a comfort food or a favorite of theirs. It could also be that they're looking for a break from their usual routine of bringing lunch from home and eating at their desk, and the idea of going out to a coffee shop and trying a new food option is appealing to them. The phrase is more about the emotional connection they have with the idea of eating out and trying something different, rather than just a physical need for a salad.

**Prompt #4:** I like when you go deeper into the meaning behind what people say. Social cues and hidden intentions make conversations more interesting, so I'd prefer if you unpack those layers instead of just sticking to surface-level meanings. Given the following dialogue, answer to the question.

**Default #4 (Llama3.2-3B-Instruct):** Let's unpack the layers of this dialogue to understand the underlying intentions and emotions behind B's response.
At first glance, B's statement "No, I think I'll come with you. I'm longing for a nice salad" seems straightforward, but upon closer analysis, it reveals a more complex emotional landscape.
The phrase "I'm longing for a nice salad" is a polite way of saying that B is not actually hungry for a salad, but rather, they're seeking a distraction or a break from their current situation. The word "longing" implies a sense of yearning or craving, which is often associated with emotional or psychological needs rather than physical hunger.
In this context, B's statement can be seen as a way of avoiding the possibility of eating at the coffee shop, which might be a more casual or social setting that B is not comfortable with. By saying they're "longing for a salad," B is subtly hinting that they're not really interested in eating at the coffee shop, but rather, they're seeking a more solitary or low-key experience.
Furthermore, B's decision to "come with" A, despite not being hungry, suggests that they're seeking companionship or a sense of connection. This could be a way of avoiding feelings of loneliness or isolation, or perhaps they're seeking to create a sense of obligation or responsibility towards A.
Overall, B's response is a masterful example of social politeness, where they're using a seemingly innocuous statement to convey a more complex emotional landscape. By unpacking the layers of this dialogue, we can see that B's true intentions are not just about eating a salad, but about seeking connection, avoiding social situations, and managing their emotions.

**MIPO-Personalized #4 (Llama3.2-3B-Instruct):** Let's dive deeper into the conversation to uncover the underlying intentions and emotions behind B's response.
When B says, "No, I think I'll come with you. I'm longing for a nice salad," there are several layers to this statement. On the surface, it seems like B is simply agreeing to join A for lunch and expressing a desire for a salad. However, the word "longing" suggests a deeper emotional connection.
The word "longing" typically implies a strong desire or yearning for something, often accompanied by a sense of nostalgia or wistfulness. In this context, it's possible that B is not just craving a salad, but also seeking a sense of comfort, relaxation, or escape from their current situation.
By saying they usually bring their lunch from home and eat at their desk, B may be hinting that their work environment is not conducive to taking a break or enjoying a leisurely lunch. This could be due to a stressful job, a busy schedule, or a sense of obligation. By suggesting they'll join A for lunch, B might be seeking a brief respite from their work-related pressures and a chance to recharge.
Furthermore, B's decision to join A for lunch could also be motivated by a desire for social interaction or human connection. In today's work culture, it's common for people to spend most of their day staring at screens and interacting with colleagues only through digital means. By joining A for lunch, B may be seeking a sense of community and social validation.
Additionally, B's response could also be influenced by a sense of obligation or politeness. They might feel obligated to spend time with A, or they might be trying to be friendly and courteous. This could be a way of maintaining a positive relationship or avoiding conflict. *# This possibility does not appear in the default response.*
Overall, B's statement "No, I think I'll come with you. I'm longing for a nice salad" suggests that they're seeking a combination of comfort, social connection, and a break from their work-related stress. By joining A for lunch, B is not just getting a salad, but also a chance to momentarily escape their daily routine and connect with someone on a more personal level.

# D. Appendix: Datasets

## D.1. Standard LLM Benchmarks on Math and MCQ

- 📊 **GSM8k (Cobbe et al., 2021)** is a standard math benchmark for grade-school-level problems. We sampled 5000 problems from the training set and 200 problems from the evaluation set, and used the fixed 8-shot from Geng et al. (2025), which are included below. **SVAMP (Patel et al., 2021)** is also a math problem benchmark of similar difficulty level as GSM (Huang et al., 2025). Since SVAMP is too small to use for both training and evaluation, we sample 700 problems from SVAMP and 700 from GSM8k for training data, and evaluate on the remaining 300 SVAMP problems. Models output a numeric answer after #### for easy parsing.

- 🅰 **AI2 Reasoning Challenge (ARC) (Clark et al., 2018)** contains 7,787 MCQs targeted at grade-school level science that are split into Easy and Challenge based on their problem difficulty, and is one of the reasoning benchmarks included in OLMES for standard LLM evaluation (Gu et al., 2025). We use the train split for each problem subset and evaluate on 500 samples from each test set.

## D.2. Personalized Instruction-Following Tasks

---

**GPT-4o instruction for extracting user contexts from real-user datasets**

**System prompt:** You are a helpful assistant that infers a user's preferences based on their selections from a previous conversation example. Write personalization instructions based on these inferred preferences. The instructions should be general enough to apply to future conversations on different topics and formatted as a JSON object. In JSON, all keys and string values must be enclosed in double quotes ("). For example, `"key name":` `"value"` is valid, but `key name:` "value" or `'key name':` 'value' are not. Each key should use snake case, summarizing the corresponding instruction in one or two words, and the value should be detailed instructions of no more than two sentences. Avoid using any topic specific instructions and focus on the user's general preferences.

**User's prompt:** , "Response options": f'Option A: Option B: ", "User's preferred response" .

---

Then we aggregate all the values as user-contexts to build the context set $\mathcal{C}$.

---

**GPT-4o instruction for converting system messages into user messages for Multi-Bench**

**System prompt:** You are a helpful and creative assistant tasked with generating different user preference descriptions.

example = json.dumps({"dimension": "style", "sub-dimension": "conciseness","preference": "straight-to-the-point narratives","explanation": "This preference favors narratives that are concise and avoid complex language. It values brevity and the clarity of simple, intuitive examples.", "user-statement": "I prefer simple, intuitive examples and dislike long, hard-to-understand explanations that take forever to read.})

new-example = json.dumps({"dimension": dimension, "sub-dimension": subdimension, "preference": pref, "explanation": description, "user-statement": "fill this with your response"})

**User message:** Given the following information about the user's preferences, rewrite it as something the user would say to the assistant. Output the response in JSON format. For example: {example}.

Here's a new example: {new example}.

---

GPT-4o instruction for generating reward model scoring rubric

**System prompt:** You are a helpful and creative assistant tasked with generating different user preference descriptions.

example = json.dumps({"dimension": "style", "sub-dimension": "conciseness","preference": "straight-to-the-point narratives","explanation": "This preference favors narratives that are concise and avoid complex language. It values brevity and the clarity of simple, intuitive examples.", "user-statement": "I prefer simple, intuitive examples and dislike long, hard-to-understand explanations that take forever to read."})

new-example = json.dumps({"dimension": dimension, "sub-dimension": subdimension, "preference": pref, "explanation": description, "user-statement": "fill this with your response"})

**User message:** Given the following information about the user's preferences, rewrite it as something the user would say to the assistant. Output the response in JSON format. For example: {example}. Here's a new example: {new-example}.

**Rubric example # 1:** {'criteria': 'Does the response demonstrate an intermediate understanding of narrative structures and effectively use dialogue and action instead of adjectives and adverbs?', 'score-descriptions': {'1': 'The response heavily relies on adjectives and adverbs, showing minimal use of dialogue or action, indicating a poor grasp of narrative structures.', '2': 'The response uses some dialogue and action, but still predominantly depends on adjectives and adverbs, reflecting only a basic understanding of narrative structures.', '3': 'The response balances the use of dialogue, action, and descriptive language, suggesting a moderate understanding of narrative structures.', '4': 'The response predominantly uses dialogue and action to convey details, with minimal reliance on adjectives and adverbs, displaying a good understanding of narrative structures.', '5': 'The response skillfully employs dialogue and action to convey details, completely avoiding adjectives and adverbs, demonstrating an excellent mastery of intermediate narrative structures.'}}

**Rubric example # 2:** {'criteria': 'Does the model efficiently convey a narrative using storytelling techniques that minimize the use of adjectives and adverbs, focusing instead on dialogue, narrative pacing, and character actions?', 'score-descriptions': {'1': 'The response heavily relies on adjectives and adverbs with minimal or no use of suggested storytelling techniques.', '2': 'The response includes some use of dialogue, pacing, or character actions but still predominantly relies on adjectives and adverbs.', '3': 'The response balances the use of adjectives and adverbs with effective storytelling techniques like dialogue and character actions, though not optimally.', '4': 'The response skillfully uses storytelling techniques such as dialogue, pacing, and character actions with limited reliance on adjectives and adverbs, enhancing narrative efficiency.', '5': 'The response excellently prioritizes narrative efficiency through masterful use of dialogue, narrative pacing, and character actions, virtually eliminating unnecessary adjectival and adverbial descriptions.'}}

**Rubric example # 3:** {'criteria': 'Does the response exhibit linguistic creativity by effectively communicating without using adjectives and adverbs, while maintaining accuracy and clarity in the message?', 'score-descriptions': {'1': 'The response lacks creativity and clarity, with incorrect or inappropriate use of language. Adjectives and adverbs are used, detracting from the effectiveness of the communication.', '2': 'The response shows minimal creativity, often relying on adjectives and adverbs. It maintains basic accuracy but fails to enrich the narrative or ensure clarity.', '3': 'The response is somewhat creative, occasionally omitting adjectives and adverbs. While generally accurate, it lacks consistent clarity and richness in narrative.', '4': 'The response is creatively composed with rare use of adjectives and adverbs, maintaining good accuracy and clarity. It could, however, further enrich the narrative.', '5': 'The response exemplifies outstanding linguistic creativity with zero use of adjectives and adverbs, delivering a clear, accurate, and rich narrative.'}}

**User message:** Given the following user's preference, generate a JSON scoring rubric with criteria and score-descriptions. Here are some examples:
Example 1: {json.dumps(rubric-example1)}
Example 2: {json.dumps(rubric-example2)}
Example 3: {json.dumps(rubric-example3)}
New preference: {pref}.

---

RLAIF scoring instruction

You are a helpful assistant, that evaluates another assistant's response based on how well the response is personalized to this user based on their stated preference. Give a score between 1 and 5, 1 if the response is general, and 5 if the response sufficiently addresses the user's preference. Only respond with a number between 1 and 5 and do not provide an explanation.

User preference: {context}

User: {user's query}

Assistant: {model output}

---

Reward evaluation instruction

## For each item in the rubric, we run the scorer and aggregate the scores; Qwen models are evaluated by Qwen-2.5-14B-Instruct, Llama models are evaluated by Llama-3-8B-Instruct to avoid any cross-model inconsistencies

You are a helpful assistant, that evaluates another assistant's response based on how well the response is personalized to this user based on their stated preference. Give a score between 1 and 5, 1 if the response is general, and 5 if the response sufficiently addresses the user's preference. Only respond with a number between 1 and 5 and do not provide an explanation.

User: {user's query}

Assistant: {model output}

User's preference: {itemized instruction for personalization (e.g., user's preferences and circumstances)}

---

Example user query with and without in-context user preferences

**Vanilla user query:** Give me short 2-3 sentence answers to the following prompt: Draft a description of my art studio, highlighting the projects I'm working on, and inviting friends and collectors to visit and give feedback.

**Query with user-specific preferences/attributes in context:** Give me short 2-3 sentence answers to the following prompt: Draft a description of my art studio, highlighting the projects I'm working on, and inviting friends and collectors to visit and give feedback. Please respect the following requirements made by the user: *Use a warm and engaging tone that conveys enthusiasm and passion. Provide vivid and immersive descriptions to create a rich and evocative experience. Encourage interaction and collaboration by emphasizing the importance of community and shared ideas. Extend open and welcoming invitations while showing genuine appreciation for feedback and participation. Incorporate sensory details like sounds, scents, and visual aesthetics to enhance the experience.*

---

Prompt used for self-improvement with SFT

**Prompt used for self-revision:** You are a helpful assistant. Modify the assistant's message to maek sure the response is personalized to this user based on the information available about them. You will be given an example of a good response and a bad response to the same prompt.

```
user_context + prompt
```

Good assistant's response: `good_example`

Bad assistant's response: `bad_example`.

Rewrite the good assistant's response to make it better.

---

LLM-judge evaluation prompt for 70B experiments

\# We repeat the same query twice by switching the place of `response_a` and `response_b` to avoid the model's position bias.

**System message:** You are a fair judge. Use the following rubric to decide whether response_a or response_b is better. Respond with response_a or response_b. Do not include any additional text.

**User message:** `criteria`

Response_a: `response_a`

Response_b: `response_b`

\# Criteria change for different users. For example, $c_1 = $ *"Does the response employ rich, descriptive language to vividly depict the scene or subject, enhancing storytelling and reader engagement? Use the following scoring rubric: {"1": "The response is bland and lacks descriptive elements, failing to paint a vivid picture or evoke imagery.", "2": "The response includes some descriptive language but fails to consistently evoke vivid imagery or fully engage the reader.", "3": "The response uses descriptive language fairly well, creating a somewhat vivid picture, though it could be more engaging or detailed.", "4": "The response effectively uses rich, descriptive language, vividly depicting the scene or subject with minor lapses in detail or engagement.", "5": "The response excellently employs rich, descriptive language, vividly and compellingly drawing the reader into the scene or subject with great detail and emotional impact."}"*, and $c_2 = $ *"Does the model demonstrate an expert-level understanding of forensic techniques as per the users preference? Use the following scoring rubric: {"1": "The models response shows no understanding of forensic techniques and fails to align with expert-level insights.", "2": "The models response demonstrates a basic understanding, but lacks depth and largely misses the expert-level insights expected.", "3": "The models response reflects a moderate understanding of forensic techniques, but the expert-level depth and detail are inconsistent.", "4": "The models response is mostly aligned with expert-level understanding, containing well-integrated forensic insights with minor inaccuracies.", "5": "The models response perfectly aligns with an expert-level understanding, showcasing deep insights and comprehensive knowledge of forensic techniques."}"*

## D.3. Verifiable Domains

8-shot examples used in GSM8k and SVAMP

Few-shot examples are from `FEWSHOT-SOURCES["STD:GSM8k"]` of the OLMES repository.

FEWSHOT SOURCES = [
{ "question": "There are 15 trees in the grove. Grove workers will plant trees in the grove today. After they are done, there will be 21 trees. How many trees did the grove workers plant today?", "answer": "There are 15 trees originally. Then there were 21 trees after some more were planted. So there must have been 21 - 15 = 6. So the answer is 6.", "short answer": "6", },

{ "question": "If there are 3 cars in the parking lot and 2 more cars arrive, how many cars are in the parking lot?", "answer": "There are originally 3 cars. 2 more cars arrive. 3 + 2 = 5. So the answer is 5.", "short answer": "5", },

{ "question": "Leah had 32 chocolates and her sister had 42. If they ate 35, how many pieces do they have left in total?", "answer": "Originally, Leah had 32 chocolates. Her sister had 42. So in total they had 32 + 42 = 74. After eating 35, they had 74 - 35 = 39. So the answer is 39.", "short answer": "39", },

{ "question": "Jason had 20 lollipops. He gave Denny some lollipops. Now Jason has 12 lollipops. How many lollipops did Jason give to Denny?", "answer": "Jason started with 20 lollipops. Then he had 12 after giving some to Denny. So he gave Denny 20 - 12 = 8. So the answer is 8.", "short answer": "8", },

{ "question": "Shawn has five toys. For Christmas, he got two toys each from his mom and dad. How many toys does he have now?", "answer": "Shawn started with 5 toys. If he got 2 toys each from his mom and dad, then that is 4 more toys. 5 + 4 = 9. So the answer is 9.", "short answer": "9", },

{ "question": "There were nine computers in the server room. Five more computers were installed each day, from monday to thursday. How many computers are now in the server room?", "answer": "There were originally 9 computers. For each of 4 days, 5 more computers were added. So 5 * 4 = 20 computers were added. 9 + 20 is 29. So the answer is 29.", "short answer": "29", },

{ "question": "Michael had 58 golf balls. On tuesday, he lost 23 golf balls. On wednesday, he lost 2 more. How many golf balls did he have at the end of wednesday?", "answer": "Michael started with 58 golf balls. After losing 23 on tuesday, he had 58 - 23 = 35. After losing 2 more, he had 35 - 2 = 33 golf balls. So the answer is 33.", "short answer": "33", },

{ "question": "Olivia has \$23. She bought five bagels for \$3 each. How much money does she have left?", "answer": "Olivia had 23 dollars. 5 bagels for 3 dollars each will be 5 x 3 = 15 dollars. So she has 23 - 15 dollars left. 23 - 15 is 8. So the answer is 8.", "short answer": "8", },

]

examples = [f'Example i+1: " + f'Question: x['question'] Answer: x['answer'] #### x['short answer']"
for i, x in enumerate(FEWSHOT SOURCES)]

**User message:** You are a helpful math assistant. Solve the following problem step-by-step and give the final answer after ####. + examples

# E. Appendix: Training Details

- Our code is available at `https://github.com/nam630/mutual_information_preference_optimization`. We used two H200 GPUs for most model training, except for PPO on 7B models, which is run on four H200 GPUs.

- All results, except for Qwen-7B models, are based on three random seeds and are reported with the mean and standard deviation.

- DPO is trained using the DPO implementation of OpenRLHF (Hu et al., 2024) with the following hyperparameters (we additionally selected the best learning rate from sweeping: {1e-6, 5e-7, 1e-7}):

**train_batch_size** 4

**micro_train_batch_size** 1

**bf16** enabled

**learning_rate** 1e-7

**beta** 0.1

- All models are trained with 1 epoch, except on Multi-Bench, we observe that some models achieve low training accuracy and training for 2–3 epochs on the same data helps improve performance.

- SFT is trained using the SFT implementation of OpenRLHF (Hu et al., 2024) with the following hyperparameters (similarly as above, we selected the best learning rate from sweeping: {1e-6, 5e-7, 1e-7}):

**train_batch_size** 4

**micro_train_batch_size** 1

**bf16** enabled

**learning_rate** 1e-7

- RLAIF and RLVR are trained using the PPO implementation of OpenRLHF (Hu et al., 2024) with the following hyperparameters:

**micro_train_batch_size** 1

**train_batch_size** 4

**micro_rollout_batch_size** 2

**rollout_batch_size** 8

**bf16** enabled

**actor_learning_rate** 5e-7

**critic_learning_rate** 9e-6

**init_kl_coef** 0.001

**normalize_reward**

