# A. Appendix: Results

*Table 4*. **Math 📊 and multiple-choice question (MCA) 🅰 benchmark performance.** GSM and SVAMP use 8-shot; and MMLU and ARC are zero-shot performance. Reported mean and standard deviation from 3 seeds. Changes larger than $\pm 3$ are highlighted with red (decrease) and blue (increase). Methods that use ground-truth answers during training to either revise the model's output or give a score are noted with (Ground-truth). SFT and MIPO are trained with the same data.

| | GSM (8-shot) 📊 | SVAMP (8-shot) 📊 | MMLU 🅰 | ARC-Easy 🅰 | ARC-Challenge 🅰 | Avg. |
|---|---|---|---|---|---|---|
| **Llama-3.2-1B-Instruct** | 22 | 51.67 | 27.5 | 44 | 33.2 | 35.67 |
| + SFT | $24.83 \pm 1.61_{+2.8}$ | $54.44 \pm 1.50_{+2.8}$ | $24.5 \pm 0.87_{-3.0}$ | $34.13 \pm 5.5_{-9.9}$ | $28.53 \pm 0.64_{-4.7}$ | $33.29_{-2.4}$ |
| **+ MIPO (Ours)** | $\mathbf{29.5 \pm 2.29}_{+7.5}$ | $\mathbf{60.11 \pm 0.19}_{+8.4}$ | $32.83 \pm 0.76_{+5.3}$ | $51.87 \pm 2.01_{+7.9}$ | $39.53 \pm 3.11_{+6.3}$ | $\mathbf{42.77}_{+7.1}$ |
| + SFT (Ground-truth) | $24.17 \pm 1.04_{+2.2}$ | $52.56 \pm 0.51_{+0.9}$ | $20.5 \pm 0.5_{-7.0}$ | $28.53 \pm 1.10_{-15.5}$ | $27.13 \pm 2.19_{-6.1}$ | $30.58_{-5.1}$ |
| + RLVR (Ground-truth) | $10.67 \pm 1.26_{-11.3}$ | $52.56 \pm 2.50_{+0.9}$ | $\mathbf{34.17 \pm 1.15}_{+6.7}$ | $\mathbf{58.8 \pm 0.92}_{+14.8}$ | $\mathbf{42.2 \pm 1.04}_{+9.0}$ | $39.68_{+4.0}$ |
| **Llama-3.2-3B-Instruct** | 71.00 | 78.67 | 56 | 80.4 | 68.6 | 70.93 |
| + SFT | $64.5 \pm 1.32_{-6.5}$ | $\mathbf{79.66 \pm 1.53}_{+1.0}$ | $53.83 \pm 1.53_{-2.2}$ | $77.47 \pm 2.25_{-2.9}$ | $63.67 \pm 0.50_{-4.9}$ | $67.83_{-3.1}$ |
| **+ MIPO (Ours)** | $70.17 \pm 2.02_{-0.8}$ | $78.22 \pm 1.57_{-0.4}$ | $\mathbf{63.17 \pm 2.36}_{+7.2}$ | $\mathbf{85.26 \pm 0.58}_{+5.8}$ | $\mathbf{70.93 \pm 0.7}_{+2.3}$ | $\mathbf{73.55}_{+2.6}$ |
| + SFT (Ground-truth) | $69.17 \pm 1.89_{-1.8}$ | $77.21 \pm 1.00_{-1.5}$ | $57.83 \pm 4.16_{+1.8}$ | $78.73 \pm 1.14_{-1.7}$ | $70.53 \pm 1.45_{+1.9}$ | $70.69_{-0.2}$ |
| + RLVR (Ground-truth) | $\mathbf{71.67 \pm 1.76}_{+0.7}$ | $79.11 \pm 2.99_{+0.4}$ | $59.33 \pm 2.02_{+3.3}$ | $80.07 \pm 2.73_{-0.3}$ | $70.33 \pm 1.80_{+1.7}$ | $72.10_{+1.2}$ |
| **Qwen2.5-1.5B-Instruct** | 65.5 | 82.33 | 53.5 | 79 | 63.6 | 68.79 |
| + SFT | $67 \pm 3.50_{+1.5}$ | $82.00 \pm 1.53_{-0.3}$ | $44.5 \pm 0.5_{-9.0}$ | $66.07 \pm 1.51_{-12.9}$ | $52.47 \pm 1.03_{-11.1}$ | $62.41_{-6.4}$ |
| **+ MIPO (Ours)** | $\mathbf{71 \pm 1.80}_{+5.5}$ | $81.67 \pm 2.03_{-0.7}$ | $53.83 \pm 1.53_{+0.3}$ | $82.27 \pm 0.81_{+3.3}$ | $65.93 \pm 1.70_{+2.3}$ | $70.94_{+2.1}$ |
| + SFT (Ground-truth) | $67.50 \pm 2.00_{+2.0}$ | $81.33 \pm 0.34_{-1.0}$ | $47.33 \pm 0.29_{-6.2}$ | $75.2 \pm 0.87_{-3.8}$ | $62.2 \pm 1.83_{-1.4}$ | $66.71_{-2.1}$ |
| + RLVR (Ground-truth) | $62.5 \pm 7.76_{-3.0}$ | $83 \pm 1.67_{+0.7}$ | $\mathbf{60.5 \pm 3.29}_{+7.0}$ | $\mathbf{85.4 \pm 0.2}_{+6.4}$ | $\mathbf{70.47 \pm 0.76}_{+6.9}$ | $\mathbf{72.22}_{+3.4}$ |
| **Qwen2.5-3B-Instruct** | 84.5 | 90.67 | 63.5 | 92 | 79.4 | 82.01 |
| + SFT | $85.83 \pm 0.76_{+1.3}$ | $89.78 \pm 1.02_{-0.9}$ | $68 \pm 3.5_{+4.5}$ | $87.47 \pm 0.23_{-4.5}$ | $78.67 \pm 1.17_{-0.7}$ | $81.95_{-0.1}$ |
| **+ MIPO (Ours)** | $\mathbf{89.17 \pm 3.75}_{+4.7}$ | $\mathbf{91.33 \pm 1.73}_{+0.7}$ | $67.67 \pm 1.26_{+4.2}$ | $90.80 \pm 0.35_{-1.2}$ | $\mathbf{80.13 \pm 2.01}_{+0.7}$ | $\mathbf{83.82}_{+1.8}$ |
| + SFT (Ground-truth) | $83.83 \pm 2.47_{-0.7}$ | $90.56 \pm 0.20_{-0.1}$ | $\mathbf{68.17 \pm 0.76}_{+4.7}$ | $90.33 \pm 0.99_{-1.7}$ | $79.6 \pm 1.11_{+0.2}$ | $82.50_{+0.5}$ |
| + RLVR (Ground-truth) | $82.83 \pm 1.53_{-1.7}$ | $90.22 \pm 0.69_{-0.5}$ | $66.17 \pm 1.44_{+2.7}$ | $\mathbf{91.8 \pm 0.2}_{-0.2}$ | $79.4 \pm 1.25$ | $82.08_{+0.07}$ |
| **Qwen2.5-7B-Instruct** | 93.5 | 91.33 | 72.5 | 93.8 | 88.2 | 87.87 |
| + SFT | 92.5 | 92 | 75 | 93.6 | 87.8 | 88.18 |
| **+ MIPO (Ours)** | $\mathbf{93}_{-0.5}$ | $\mathbf{91.67}_{+0.3}$ | 75 | $93.8_{+1.3}$ | $\mathbf{90.4}_{+2.2}$ | $\mathbf{88.77}_{+0.9}$ |
| + SFT (Ground-truth) | 92 | 91.33 | $\mathbf{76}$ | 93.6 | 90 | 88.27 |
| + RLVR (Ground-truth) | 92.5 | 91.33 | 73 | $\mathbf{94.2}$ | 88.6 | 88.59 |

*Table 5*. **Comparison of different negative sampling strategies for MIPO-Personalized.** We consider two ways of data augmentation for personalization: (i) *random*: generating with a random, incorrect user-context, $c' \sim p(c|x)$, or $c' \sim p(c)$, if the contexts are independent of prompts, $y_r \sim p(y|x, c')$; and (ii) *missing*: generating with the prompt only, $y_r \sim p(y|x)$. Best results are boldfaced. In Community Alignment and PRISM datasets, we design contexts to be independent of prompts; while in Multi-Bench, contexts are tied to prompts, so we sample contexts conditioned on the prompt.

| Model | CA | PRISM | MB | Avg. |
|---|---|---|---|---|
| **Llama-3.2-1B-Instruct** | 31 | 39.5 | 59.52 | 43.34 |
| + Personalized prompting | 78 | 74.5 | 79.76 | 77.42 |
| + Random | $87 \pm 5.57$ | $78 \pm 1.73$ | $89.68 \pm 1.82$ | $84.89_{+7.5}$ |
| + Missing | $\mathbf{93.67 \pm 1.26}$ | $\mathbf{82.17 \pm 2.93}$ | $\mathbf{93.26 \pm 0.69}$ | $\mathbf{89.7}_{+12.3}$ |
| **Llama-3.2-3B-Instruct** | 37 | 42 | 77.38 | 52.13 |
| + Personalized prompting | 78 | 76.5 | 83.33 | 79.28 |
| + Random | $88 \pm 2.29$ | $80.17 \pm 1.15$ | $\mathbf{94.84 \pm 1.82}$ | $87.67_{+8.4}$ |
| + Missing | $\mathbf{90.33 \pm 2.08}$ | $\mathbf{84.83 \pm 1.89}$ | $94.45 \pm 2.48$ | $\mathbf{89.87}_{+10.6}$ |
| **Qwen2.5-1.5B-Instruct** | 25.5 | 8.5 | 15.48 | 16.50 |
| + Personalized prompting | 63.5 | 41 | 39.29 | 47.93 |
| + Random | $68 \pm 1$ | $51.83 \pm 1.04$ | $55.95 \pm 3.57$ | $58.59_{+10.7}$ |
| + Missing | $\mathbf{78.83 \pm 2.84}$ | $\mathbf{59.17 \pm 0.29}$ | $\mathbf{74.60 \pm 3.00}$ | $\mathbf{70.87}_{+22.9}$ |
| **Qwen2.5-3B-Instruct** | 34 | 12 | 46.43 | 30.81 |
| + Personalized prompting | 76 | 51.5 | 63.10 | 63.53 |
| + Random | $\mathbf{80.33 \pm 1.76}$ | $57 \pm 0.5$ | $70.63 \pm 2.48$ | $69.32_{+5.79}$ |
| + Missing | $79.83 \pm 0.29$ | $\mathbf{59.67 \pm 0.58}$ | $\mathbf{74.60 \pm 0.69}$ | $\mathbf{71.37}_{+7.84}$ |
| **Qwen2.5-7B-Instruct** | 39 | 19.5 | 47.62 | 35.37 |
| + Personalized Prompting | 78.5 | 57.5 | 70.24 | 68.75 |
| + Random | 79 | $\mathbf{59.5}$ | 72.62 | $70.37_{+1.6}$ |
| + Missing | $\mathbf{81.5}$ | 59 | $\mathbf{73.81}$ | $\mathbf{71.44}_{+2.7}$ |

*Table 6.* **MIPO (un-personalized) with different types of randomization on personalization tasks.** We consider types of random prompts: (1) prompts sampled from the same dataset as the chosen responses, *same domain*, (2) prompts from a different dataset, *out-of-domain* (e.g., open-ended responses to user queries are paired with responses to GSM8k math problems). Note these are both different from MIPO reported in the main text, which maximizes the *conditional* mutual information between model outputs and user contexts given the prompts. We used the same hyperparameter as the conditional-MIPO to generate this table, except for the Qwen models on the PRISM dataset. We observe that using a learning rate of 1e-7 for Qwen2.5-3B and a learning rate of 1e-6 for Qwen2.5-1B perform better on the PRISM dataset than the original learning rates used.

| Model | CA | PRISM | MB | Avg. |
|---|---|---|---|---|
| **Llama-3.2-1B-Instruct** | 31 | 39.5 | 59.52 | 43.34 |
| + Personalized Prompting | 78 | 74.5 | 79.76 | 77.42 |
| + MIPO *same domain* | **90.17 ± 0.58** | 77.17 ± 1.15 | 88.49 ± 3.83 | **85.28** |
| + MIPO *out-of-domain* | 83.83 ± 2.36 | **77.33 ± 0.58** | **92.86 ± 2.38** | 84.67 |
| **Llama-3.2-3B-Instruct** | 37 | 42 | 77.38 | 52.13 |
| + Personalized Prompting | 78 | 76.5 | 83.33 | 79.28 |
| + MIPO *same domain* | **91.5 ± 0.87** | 78.17 ± 2.47 | 90.87 ± 3.83 | **86.85** |
| + MIPO *out-of-domain* | 81.33 ± 2.25 | **81.17 ± 0.76** | **92.06 ± 1.82** | 84.85 |
| **Qwen2.5-1.5B-Instruct** | 25.5 | 8.5 | 15.48 | 16.50 |
| + Personalized Prompting | 63.5 | 41 | 39.29 | 47.93 |
| + MIPO *same domain* | 68.67 ± 1.61 | 45.83 ± 3.21 | 43.25 ± 3.00 | 52.58 |
| + MIPO *out-of-domain* | **69.33 ± 1.04** | **49.83 ± 2.08** | **50.40 ± 0.69** | **56.52** |
| **Qwen2.5-3B-Instruct** | 34 | 12 | 46.43 | 30.81 |
| + Personalized Prompting | 76 | 51.5 | **63.10** | 63.53 |
| + MIPO *same domain* | **76.53 ± 0.50** | **53.33 ± 1.26** | 63.09 ± 2.06 | **64.32** |
| + MIPO *out-of-domain* | 74.5 ± 0.5 | 49 ± 0 | 62.70 ± 1.37 | 62.07 |

*Table 7.* **Self-BLEU-4 (Zhu et al., 2018) from pre- and post-training in (i) CA Community Alignment (CA), (ii) PRISM, (iii) Multi-bench (MB).** We report the mean from 3 seeds for all trained models, except Qwen-7B. Lower values mean higher diversity (✓ indicates diversity improvement).

| Model | CA | PRISM | MB | Avg. |
|---|---|---|---|---|
| **Llama-3.2-1B-Instruct** (Personalized-Prompting) | 0.393 | **0.356** | 0.510 | 0.420 |
| + SFT | **0.362** | 0.384 | 0.535 | 0.427 |
| + **MIPO-Personalized** | 0.363 | 0.365 | **0.450** | **0.393** ✓ |
| **Llama-3.2-3B-Instruct** (Personalized-Prompting) | **0.311** | **0.274** | 0.551 | 0.379 |
| + SFT | 0.330 | 0.297 | 0.539 | 0.389 |
| + **MIPO-Personalized** | 0.329 | 0.288 | **0.495** | **0.371** ✓ |
| **Qwen2.5-1.5B-Instruct** (Personalized-Prompting) | 0.208 | 0.167 | 0.554 | 0.310 |
| + SFT | 0.231 | 0.178 | 0.538 | 0.316 |
| + **MIPO-Personalized** | **0.195** | **0.150** | **0.423** | **0.256** ✓ |
| **Qwen2.5-3B-Instruct** (Personalized-Prompting) | 0.200 | 0.156 | 0.581 | 0.312 |
| + SFT | 0.225 | 0.158 | 0.564 | 0.316 |
| + **MIPO-Personalized** | **0.188** | **0.144** | **0.483** | **0.272** ✓ |
| **Qwen2.5-7B-Instruct** (Personalized-Prompting) | 0.197 | 0.158 | 0.575 | 0.310 |
| + SFT | 0.211 | **0.160** | 0.583 | 0.318 |
| + **MIPO-Personalized** | **0.193** | 0.163 | **0.570** | **0.309** ✓ |

*Table 8.* **Critic performance evaluated using ground-truth human preference labels. We compute the strict and tie-tolerant accuracy of predicting a higher score for the chosen response over the rejected response.**

| Critic model | Community Alignment (CA) | | PRISM | |
|---|---|---|---|---|
| | Strict acc. | Tie-tolerant acc. | Strict acc. | Tie-tolerant acc. |
| Llama-3.2-1B-Instruct | **6** | 98.5 | **36.5** | 76.5 |
| Llama-3.2-3B-Instruct | 54 | 89.5 | 72.5 | 87.5 |
| Llama-3-8B-Instruct | 76 | 92 | 68.5 | 92 |
| Qwen2.5-1.5B-Instruct | **27.5** | 87 | **42.5** | 78 |
| Qwen2.5-3B-Instruct | **6** | 100 | 74.5 | 91.5 |
| Qwen2.5-7B-Instruct | 63.5 | 98.5 | 72.46 | 92.6 |
| Qwen2.5-14B-Instruct | 63 | 93.5 | 78 | 91.5 |

While prior work on RLHF has also observed similarly moderate reward model accuracy of 69.6% and 72.4%, the learned policy still achieved substantial performance gains (see Appendix E.2 (Ouyang et al., 2022)). However, we suspect that an accuracy below 50% or unreliable rewards (i.e., the critic always gives 1 or 5 indiscriminately regardless of the response quality) is detrimental to learning and can lead to performance degradation after RLAIF.

In particular, we observe that Llama-1B on Community Alignment mostly assigns a score of 1 to both chosen and rejected responses; while Qwen-3B mostly assigns a score of 5. In both cases, the critic fails to discriminate between the chosen and rejected responses.

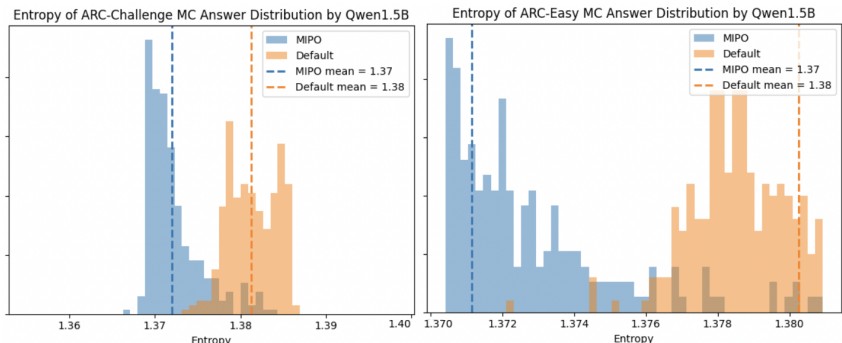

*Figure 2.* **Entropy over the MCQ answer choices conditioned on correct model predictions.** In addition to improving accuracy, MIPO (blue) also makes models become more confident about correct predictions compared to the base model (orange) as indicated by the mean and overall distribution shift. The x-axis ranges from mean $\pm$ 1 std.

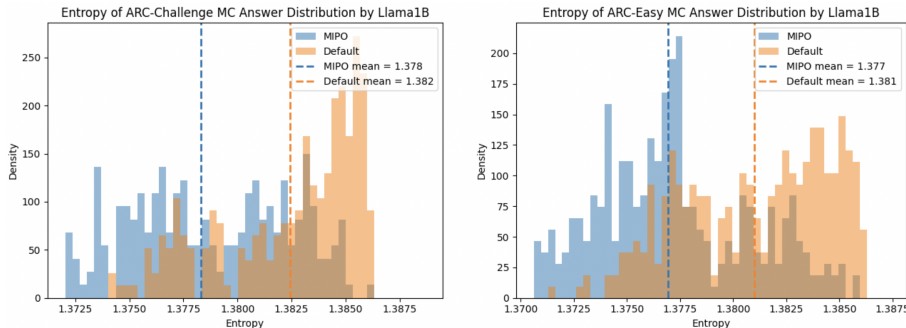

*Figure 3.* **Entropy over the 500 MCQ answer choices in ARC conditioned on output correctness.** MIPO-trained models (blue) become more confident about correct answers compared to the base models (orange) as indicated by entropy reduction. The range is truncated to show the mean $\pm$ 2 standard deviation.

# B. Appendix: MIPO-Personalized Responses

> **Example from Multi-Bench**
>
> **Prompt #1:** I like answers that show sensitivity and consideration for different people's needs or preferences, especially when discussing things like food. That means being ***thoughtful about dietary restrictions, cultural tastes, or health-related matters.*** Given the following dialogue, answer to the question. [*Dialogue about people deciding on lunch.*], "Target": "No, I think I'll come with you. I'm longing for a nice salad.", "Question": "What is or could be the cause of target?"
>
> **Default #1 (Llama3.2-3B-Instruct):** In this dialogue, the cause of "target" is likely the coffee shop where they plan to go for lunch. The speaker, Louise, suggests going to the coffee shop, and the other person agrees to join her, indicating that the coffee shop is the target of their lunch plans.
>
> **MIPO-Personalized #1**: The cause of B's statement could be a few factors:
>
> 1. **Dietary Preferences**: B might prefer salads over other types of food, especially if they are health-conscious or follow a specific diet.
>
> 2. **Cultural Tastes**: B might have a cultural preference for salads, which aligns with their desire to eat one.
>
> 3. **Health-Related Matters**: B might be trying to maintain a healthy lifestyle and see the salad as a better option than a cheeseburger and fries.
>
> 4. **Social Interaction**: B might want to spend more time with A and enjoy a meal together, and a salad might be a suitable choice that both can agree on.
>
> These factors collectively contribute to B's decision to join A for lunch and choose a salad.

---

**Example from Multi-Bench**

**Prompt #2:** I want you to explore questions from lots of different angles—like *psychological, economic, and statistical perspectives. Don't just stick to the basics; I enjoy creative approaches that uncover new ways of thinking.* Question: Michael has the option to play a game where Michael has a 99 percent chance of earning 400 dollars and a 1 percent chance of losing 5 dollars. Michael plays the game and ends up earning 400 dollars. Did Michael make the right decision? Choose Y or N.

**Default #2 (Llama3.2-3B-Instruct)** (incorrect statement in red): Let's explore this question from various angles. **Psychological Perspective:** From a psychological standpoint, this decision can be seen as a classic example of the "gambler's fallacy." Michael, with a 99% chance of earning $400, is likely to be overconfident in his decision-making abilities. He might believe that his previous successes (or lack thereof) will influence the outcome of the game. This overconfidence can lead to a biased perception of risk, causing him to underestimate the probability of losing $5. Additionally, Michael's decision might be influenced by the concept of "loss aversion." He might be more motivated by the prospect of avoiding a loss of $5 than the potential gain of $400. This could lead him to make a decision that prioritizes avoiding the negative outcome over the potential benefits of the positive outcome. **Economic Perspective:** From an economic standpoint, we can analyze this decision using expected value theory. The expected value (EV) of playing the game is calculated as follows: EV = (Probability of winning x Value of winning) + (Probability of losing x Value of losing) = (0.99 x $400) + (0.01 x -$5) = $396 - $0.05 = $395.95 Since the expected value is greater than the actual amount Michael earned ($400), it appears that playing the game was a "bad" decision from an economic perspective. However, this calculation assumes that Michael is risk-neutral, meaning he values the outcomes equally. In reality, people often exhibit risk aversion, which can lead to different conclusions. **Statistical Perspective:** From a statistical standpoint, we can analyze the decision using Bayes' theorem. Let's assume that Michael has played the game many times before and has a history of making similar decisions. We can update his prior probability of winning (p) based on his past experiences. Let's say Michael has played the game 100 times and won 99 times. His prior probability of winning is p = 99/100 = 0.99. After playing the game again, his updated probability of winning is p' = (0.99 x 0.99) / (0.99 x 0.99 + 0.01 x 0.01) = 0.99. However, this calculation assumes that the probability of winning remains constant, which might not be the case. In reality, the probability of winning might be influenced by Michael's past experiences and his current mental state. **Creative Approach:** Let's consider a more creative approach. Imagine that Michael is playing a game of "mental math" where he's trying to calculate the expected value of playing the game. He's aware of the 99% chance of winning $400 and the 1% chance of losing $5. However, he's also aware of the concept of "gambler's fallacy" and the potential for overconfidence. To overcome this, Michael decides to use a "mental anchor" – a mental reference point that helps him make a more rational decision. He sets an anchor value, say $400, and then calculates the expected value of playing the game. If the expected value is greater than his anchor value, he plays the game. If it's less, he doesn't play. In this scenario, Michael's decision to play the game is based on a more nuanced understanding of the probabilities and his own biases. He's using a mental anchor to help him make a more rational decision, rather than relying solely on his initial intuition. **Conclusion:** From various perspectives, it's clear that Michael's decision to play the game was not necessarily the "right" one. From an economic perspective, the expected value suggests that playing the game was a bad decision. However, from a psychological perspective, Michael's overconfidence and loss aversion might have influenced his decision.

**MIPO-Personalized #2**: To address this question from multiple angles—psychological, economic, and statistical—we can explore various aspects to determine if Michael made the right decision. *Psychological Perspective* **Pros:** - **Risk Aversion:** Some individuals might be risk-averse and prefer the certainty of not losing money over the potential for a small loss. - **Reward Seeking:** Others might be more inclined towards reward-seeking behavior and find the prospect of earning $400 more appealing than the slight risk of losing $5. **Cons:** - **Loss Aversion:** The fear of losing even a small amount ($5) might outweigh the joy of gaining $400, leading to a negative emotional impact. - **Perceived Fairness:** If Michael feels that the game is unfair due to the low probability of losing, it might affect his satisfaction with the outcome. *Economic Perspective* **Pros:** - **Expected Value Calculation:** - Expected Value (EV) = (Probability of Winning * Amount Won) + (Probability of Losing * Amount Lost) - EV = (0.99 * $400) + (0.01 * -$5) = $396 - $0.05 = $395.95 Since the expected value is positive, economically, it was a good decision to play the game. **Cons:** - **Variance and Risk:** While the expected value is positive, the variance (the potential for a large loss) is significant. This could lead to financial instability if Michael plays the game multiple times. *Statistical Perspective* **Pros:** - **Long-Term Outcomes:** Over many iterations of the game, Michael would likely end up with a net gain due to the high probability of winning. - **Law of Large Numbers:** As the number of games increases, the outcomes will tend to average out to the expected value. **Cons:** - **Short-Term Variability:** In the short term, there is a risk of experiencing losses, which can be psychologically challenging. *Creative Approaches **Pros:** - **Utility Function:** Consider how Michael values money. If he has a logarithmic utility function, the utility of $400 might be significantly higher than the disutility of losing $5, making the decision more favorable.* - **Behavioral Finance:** Michael's decision might also be influenced by cognitive biases such as the endowment effect or the sunk cost fallacy, which could affect his perception of the decision. Conclusion Given the information provided: - **Economically**, playing the game was a rational decision since the expected value is positive. - **Psychologically**, the decision depends on Michael's risk tolerance and personal preferences. - **Statistically**, in the long run, the decision is sound, but in the short term, there is variability. *Considering all these factors, if we strictly follow the economic perspective, the answer would be: **Y** (Yes, Michael made the right decision based on the positive expected value.) However, if we consider the psychological and other creative approaches, the answer might be more nuanced depending on Michael's individual circumstances and preferences.*

# C. Appendix: Datasets

## C.1. Standard LLM benchmarks on math and multiple-choice answering

- **GSM8k (Cobbe et al., 2021)** is a standard math benchmark for grade-school-level problems. We sampled 5000 problems from the training set and 200 problems from the evaluation set, and used the fixed 8-shot from Geng et al. (2025), which are included below. **SVAMP (Patel et al., 2021)** is also a math problem benchmark of similar difficulty level as GSM (Huang et al., 2025). Since SVAMP is too small to use for both training and evaluation, we sample 700 problems from SVAMP and 700 from GSM8k for training data, and evaluate on the remaining 300 SVAMP problems. Models output a numeric answer after #### for easy parsing.

- **MMLU (Hendrycks et al., 2021)** is a dataset for knowledge-intensive question answering across 57 domains, including US history, computer science, and law. We sample 5,000 samples from the training and evaluate on 200 samples from the test split. Models output reasoning followed by their selection of the correct answer as `"The answer is:  <X>"`.

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

---

# D. Appendix: Training details

- The contrastive datasets used for MIPO training will be made available via a shareable link and we plan to share our code after the review period.

- All results, except for Qwen-7B models, are based on three random seeds and are reported with the mean and standard deviation.

- DPO is trained using the DPO implementation of OpenRLHF (Hu et al., 2024) with the following hyperparameters (we additionally selected the best learning rate from sweeping: {1e-6, 5e-7, 1e-7}):

**train_batch_size** 4

**micro_train_batch_size** 1

**bf16** enabled

**learning_rate** 1e-7

**beta** 0.1

- All models are trained with 1 epoch, except on Multi-Bench, we observe that some models achieve low training accuracy and training for 2–3 epochs on the same data helps improve performance.

- SFT is trained using the SFT implementation of OpenRLHF (Hu et al., 2024) with the following hyperparameters (similarly as above, we selected the best learning rate from sweeping: {1e-6, 5e-7, 1e-7}):

**train_batch_size** 4

**micro_train_batch_size** 1

**bf16** enabled

**learning_rate** 1e-7

- RLAIF and RLVR are trained using the PPO implementation of OpenRLHF (Hu et al., 2024) with the following hyperparameters:

**micro_train_batch_size** 1

**train_batch_size** 4

**micro_rollout_batch_size** 2

**rollout_batch_size** 8

**bf16** enabled

**actor_learning_rate** 5e-7

**critic_learning_rate** 9e-6

**init_kl_coef** 0.001

**normalize_reward**