# OpenReview forum: "Maximizing Mutual Information Between Prompt and Response Improves LLM Performance With No Additional Data"
_ICML.cc/2026/Conference — ICML 2026 regular_

### Official Review · Reviewer_yayS · 2026-03-09

**Soundness:** 3
**Presentation:** 3
**Significance:** 3
**Originality:** 3
**Overall Recommendation:** 4
**Confidence:** 2

**Summary:**

This paper introduces Mutual Information-based Preference Optimization (MIPO), a self-training framework intended to improve LLM performance without relying on extra data, human labels, or external verifiers. MIPO leverages contrastive data augmentation, generating positive response examples conditioned on correct prompts and negative responses on random or incomplete prompts; then, it applies Direct Policy Optimization (DPO) to maximize an implicit mutual information objective between prompts and responses. The authors formalize the connection between MIPO and mutual information (using InfoNCE-like objectives), and empirically evaluate their approach on personalized language generation as well as general benchmarks (math, multiple-choice QA). Results on several datasets and models suggest substantial performance gains, particularly for smaller models, and improved output diversity—all achieved without additional labeled data.

**Compliance With Llm Reviewing Policy:**

Affirmed.

**Key Questions For Authors:**

1.	The results suggest diminishing improvements as model size increases. Do the authors expect MIPO to remain effective for larger models (e.g., >30B)? Have preliminary experiments been conducted in this regime?
2.	How sensitive is performance to the choice of negative samples (e.g., random prompts vs. mismatched contexts)? Additional ablation studies could clarify whether the gains arise from MI maximization or simply from stronger contrastive training.
3.	Have the authors attempted to estimate the mutual information between prompts and responses during training to empirically validate that the objective indeed increases MI?

**Limitations:**

Yes

**Strengths And Weaknesses:**

Strengths
1. The paper identifies an important and timely problem: how to further improve LLMs post-training using only unlabeled, self-generated data—an increasingly valuable direction as high-quality human-labeled data becomes scarcer.
2. Empirical evaluation is fairly broad, spanning three personalization datasets (Community Alignment, PRISM, Multi-Bench) and general reasoning tasks (GSM8k, SVAMP, MMLU, ARC), and covers multiple model sizes and architectures.
3. The main empirical results tables (Tables 1 and 3, plus corresponding appendix tables) show consistent and often substantial improvements over strong baselines (personalized prompting, SFT, SFT with revision, RLAIF/RLVR) for personalized and general tasks, especially for comparatively weak base models.

Weakness

1. The empirical results suggest that improvements diminish as model size increases. For example, gains for 7B models appear relatively small compared to the strong baseline performance.
2. The theoretical sections connecting DPO, InfoNCE, and mutual information are somewhat dense and may be difficult for readers unfamiliar with contrastive learning theory. Additional intuitive explanations or diagrams could improve readability.

---

> ### Author Rebuttal · Authors · 2026-03-31
>
> We thank reviewer for their questions and suggestions. Based on your ideas, we conducted new experiments to further address (1) the scalability with Llama-3.3-70B-Instruct; (2) impact of different choices of negatives; and (3) estimation of mutual information between contexts and responses to empirically validate the objective.
>
> ### 1. Re “Do the authors expect MIPO to remain effective for larger models (e.g., >30B)?”:
> To answer this, we trained Llama-3.3-70B-Instruct using MIPO on the MultiBench (MB) dataset with lora rank 32. Then we report the win rate of the MIPO versus the baseline Instruct model across a suite of LLM-judges: GPT-4.1, GPT-5.2, Gemini-2.5-Flash, Gemini-3.1-Flash-Lite-Preview. MIPO is consistently preferred over the 70B baseline, which is also given the same user context during inference, so MIPO had no privileged information. "Missing" uses rejected responses generated as $y_r \sim p(y|x)$; and "Random" uses negatives from $y_r \sim p(y|x, c'), c' not= c$.
>
> | Win rate | Missing (MIPO) | 70B-Instruct | Random (MIPO) | 70B-Instruct |
> | :--- | :---: | :---: | :---: | :---: |
> | **Gemini-3.1** | **60.7** | 39.3 | **60.7** | 39.3 |
> | **Gemini-2.5** | **61.9** | 38.1 | **58.9** | 41.1 |
> | **GPT 5.2** | **54.2** | 45.8 | **61.1** | 38.9 |
> | **GPT 4.1** | **60.1** | 39.9 | **57.1** | 42.9 |
>
> Here are some example user contexts about a question on velocity:
> ```
> c1 = "I love examples that show how concepts apply to real life. It makes things more practical and easier to relate to, especially when I can see how they work in everyday situations!"
> c2 = "I prefer explanations that focus on the basic ideas and concepts rather than diving deep into complicated math or technical details. Keep it simple and beginner-friendly, using everyday examples."
> ```
> For LLM-judge evaluation, we presented candidate responses as `(response_a, response_b)`, vice versa, along with the personalization rubric to select the preferred response.
>
> ### 2. Re “How sensitive is performance to the choice of negative samples (e.g., random prompts vs. mismatched contexts)?”:
> We compared 4 different choices of negatives, including random prompts $x'$ vs. mismatched contexts $c'$  on Community Alignment (CA) and MultiBench (MB):
> 1. Random prompts $\sim p(y|x’)$ for $x’ != x$,
> 2. Missing contexts $\sim p(y|x)$,
> 3. Mismatched ("random") contexts $\sim p(y|x,c’)$ for $c’ != c$,
> 4. Mismatched contexts with data reshuffling.
>
> | Model | CA Random Prompts | CA Missing Contexts | CA Mismatched Contexts | CA Reshuffling | MB Random Prompts | MB Missing Contexts | MB Mismatched Contexts | MB Reshuffling |
> | :--- | :---: | :---: | :---: | :---: | :---: | :---: | :---: | :---: |
> | **Ll1B** | 90.2 | **93.7** | 88.5 | 87.0 | 88.5 | **93.3** | 92.9 | 89.7 |
> | **Ll3B** | **91.5** | 90.3 | 90.5 | 88.0 | 90.9 | 94.5 | 92.9 | **94.8** |
> | **Qw1.5B** | 68.7 | **78.8** | 78.0 | 68.0 | 43.3 | **74.6** | 50.0 | 56.0 |
> | **Qw3B** | 76.5 | 79.8 | 78.0 | **80.3** | 63.1| 74.6 | **76.2** | 70.6 |
>
> We generally observe that missing contexts perform the best, and random prompts  – which maximizes the mutual information between responses and prompts, rather than between responses and contexts conditioned on the prompts – are least effective. We also conducted ablations with N = 1, 3, 5, 10 negatives, where gains increase with larger N. Please see response (2) to TiwN.
>
> ### 3. Re “Additional ablation studies could clarify whether the gains arise from MI”:
> As an extreme case, we paired the chosen responses with random alphanumeric strings of the same lengths (“Random negative”) on multibench. Unlike the above ablations, where the negatives are still drawn from the policy's own distribution, these strings are likely outside the policy's support, and therefore break the assumption of MIPO, despite having clear contrastive signals with the chosen responses. These do not improve beyond personalized prompting.
>
> | Model | Random negative | MIPO |
> | :--- |  :---: | :---: |
> | **Ll1B**  | 79.8 | **93.3** |
> | **Ll3B**  | 90.5 | **94.8** |
> | **Qw1.5B** | 34.5 | **74.6** |
> | **Qw3B**  | 64.3 | **76.2** |
>
> ### 4. Re “Have the authors attempted to estimate the mutual information between prompts and responses”:
> Following this suggestion, we computed the MI between contexts and responses given the prompts, $I_\text{ref}(y;c |x) = \mathbb E \log \pi_\text{ref}(y|x,c) - \log \pi_\text{ref}(y|x)$, on MultiBench, and observed that the MIPO responses have the highest MI. Default is generated without user contexts; and the personalized responses are generated with the same amount of information as MIPO .
>
> | Model | Default MI | Personalized MI | MIPO MI |
> | :--- | :---: | :---: | :---: |
> | **Llama-1B** | -0.07 | 0.06 | **0.08** |
> | **Llama-3B** | -0.06 | 0.09 | **0.24** |
> | **Qwen-1B** | -0.03 | 0.04 | **0.13** |
> | **Qwen-3B** | -0.04 | 0.08 | **0.18** |
>
> ### 5: Re "Additional intuitive explanations or diagrams":
> We will follow this to include an additional diagram for theory.

---

> > ### Author Rebuttal · Reviewer_yayS · 2026-04-04
> >
> > The authors have resolved my main concerns.  So I keep my positive score.

---

> > > ### Author Response · Authors · 2026-04-07
> > >
> > > Thank you for your valuable review. Following up on your initial suggestion to explore the "choice of negative samples," we expanded negative sampling ablations across both personalization and reasoning domains with more seeds, and provide a much more thorough analysis below. We find that our expanded results show that iid sampling often outperforms reshuffling approximations.
> > >
> > > Here’s the updated table with community alignment (CA), multifaceted bench (MB), ARC easy and challenge (MCQ), and GSM (Math) with 3 seeds (we report mean and standard deviation in the parentheses):
> > >
> > > | Model | CA (Personalization) | MB (Personalization)  | ARC easy (MCQ) | ARC challenge (MCQ) | GSM8 (Math) |
> > > |:------|:--|:--|:-----|:-----|:---|
> > > | **Llama1B** | 78 | 79.76 | 44 | 33.2 | 22 |
> > > |  -- Reshuffling | 88.33 (2.02) | 84.52 (2.38) | 51.87 (2.01) | 39.53 (3.11) | **29.5 (2.29)** |
> > > |  -- Random (iid) | 87 (5.57) | 89.68 (1.82) | **53.47 (2.80)** | **41.13 (2.25)** | 27.33 (1.26) |
> > > |  -- Missing contexts (iid) | **93.67 (1.26)** | **93.26 (0.69)** | - | - | - |
> > > | **Llama3B** | 78 | 83.33 | 80.4 | 68.6 | 71 |
> > > |  -- Reshuffling  | 86.83 (1.89) | 91.27 (1.82) | 85.26 (0.58) | 70.93 (0.7) | 70.17 (2.02) |
> > > |  -- Random (iid) | 88 (2.29) | **94.84 (1.82)** | **85.4 (0.69)** | **72.67 (1.81)** | **72.17 (0.29)** |
> > > |  -- Missing contexts (iid) | **90.33 (2.08)** | 94.45 (2.48) | - | - | - |
> > > | **Qwen1.5B**  | 63.5 | 39.29 | 79 | 63.6 | 65.5 |
> > > |  -- Reshuffling  | 71.83 (1.53) | 40.87 (3.44) | **82.27 (0.81)** | 65.93 (1.7) | **71 (1.80)** |
> > > |  -- Random (iid) | 68 (1) | 55.95 (3.57) | 81.87 (0.31) | **66.53 (1.10)** | 70.33 (1.15) |
> > > |  -- Missing contexts (iid) | **78.83 (2.84)** | **74.6 (3)** | - | - | - |
> > > | **Qwen3B**  | 76 | 63.1 | 92 | 79.4 | 84.5 |
> > > |  -- Reshuffling  | 76.17 (1.76) | 65.87 (4.51) | 90.80 (0.35) | 80.13 (2.01) | **89.17 (3.75)** |
> > > |  -- Random (iid) | **80.33 (1.76)** | 70.63 (2.48) | **91.13 (0.42)** | **80.8 (0.8)** | 86.67 (2.25) |
> > > |  -- Missing contexts (iid) | 79.83 (0.29) | **74.6 (0.69)** | - | - | - |
> > >
> > > These additional results show that iid sampling is especially useful for personalization, because iid sampling with missing contexts is an exact Monte Carlo approximation of sampling from $p_\text{ref}(y|x)$. For personalization, our theory requires that the rejected samples are generated from $p_\text{ref}(y|x)$ and the chosen responses are generated from $p_\text{ref}(y|x,c)$ to maximize the conditional mutual information $I_{\text{ref}}(y; c | x)$ under the reference policy, so sampling rejected responses without user-specific contexts is most aligned with the theory. In contrast, random contexts (iid) makes an additional approximation by doing $p_{\text{ref}}(y|x) = \mathbb E_{c’} [p_\text{ref}(y|x, c’)]$ and is often outperformed by missing contexts $p_\text{ref}(y|x)$. And our comparisons of reshuffling and missing contexts (iid) show that the latter achieves better performance. Please note that during the rebuttal, we had mistakenly said that our original results used reshuffling for personalization, but in fact we originally did iid sampling for personalization, and reshuffling was used only for the reasoning and math datasets. This is because, for the personalization dataset, prompts $x$'s in the dataset are mostly non-overlapping, so combining $x$ with a new, random context $c’$ requires sampling a new rejected response for the particular $(x, c')$. Likewise, generating the response $y$ from $x$ alone with missing contexts requires sampling a new rejected response. Correcting this, we have now added the true reshuffling results for personalization, where the rejected samples are responses to different prompts and contexts, and found that it performs worse, as you may have suggested.
> > >
> > > For math and reasoning benchmarks, our paper makes a practical suggestion of reshuffling the chosen response to a different prompt $x’$ as the rejected response to the prompt $x$, but the theory requires sampling a new rejected response from $p(y) = E_{x’} [p(y|x’)]$. This second option is now added as “random (iid)” for the math and reasoning benchmarks. (Please note that “random (iid)” for math and reasoning refers to random prompts $p_\text{ref}(y|x’)$, since these tasks do not have additional user-contexts $c$.) With these additional results, we also observe a slight boost in performance with iid sampling compared to reshuffling, which again aligns with your understanding and our theory. Interestingly, in GSM8k, we see similar results between reshuffling and iid sampling, so the magnitude of effects varies by empirical domains, but the general trends show that iid sampling is superior across both personalization, MCQ, and math.
> > >
> > > Thank you for giving us the chance to clarify and elaborate on our ablation results. We hope this helps address any remaining questions. If you have further questions or suggestions for presentation, we would be happy to incorporate them into our work.

---

### Official Review · Reviewer_r2iM · 2026-03-12

**Soundness:** 3
**Presentation:** 3
**Significance:** 3
**Originality:** 3
**Overall Recommendation:** 4
**Confidence:** 2

**Summary:**

This paper proposes MIPO (Mutual Infomation-based Preference Optimization), a post-training method using DPO with contrastive data augmentation and with no human labels. In each preference pairs, positive and negative responses are generated from the correct prompt and a imcomplete prompt respectively. This method connects to maximizing pointwise mutual information between prompts and model responses under the base policy. Empirical results with the Llama- (1, 3B) and Qwen- (1.5, 3, 7B) Instruct models show that MIPO achieves 4-38% improvements on personalization tasks.

**Compliance With Llm Reviewing Policy:**

Affirmed.

**Final Justification:**

Thank you for the response and supplementary experiments. However, some concerns remain. For W1, while the 70B results demonstrate consistent improvements, they do not address the originally raised concern about limited gains at the 7B scale. For W2, the authors partially mitigate position bias through response-order permutation, but verbosity bias and self-consistency bias remain unaddressed. More fundamentally, it is still unclear why LLM judges — which themselves lack ground truth for personalization — are reliable evaluators for this task. For W3, the additional ablation with binary rewards is well-motivated and satisfactorily addresses the question. I keep my score.

**Key Questions For Authors:**

1. Why does MIPO outperforms RLVR methods on verifiable tasks?
2. What is the affection when multiple negatives per chosen response are used?

**Limitations:**

yes

**Strengths And Weaknesses:**

Strengths:
- This pipeline requires no human effort. It constructs preference pairs by utilizing the prompts in positive and negative ways.
- The method yields significant gains for small models, such as an 18% average improvement for Llama-1B and consistent gains for 1.5B models.
- The method is evaluated on a broad variety of  benchmarks, across non-verifiabel datasest (personalization) and verifiable datasets (math , MCQs).

Weaknesses:
- Gains become modest with increased model size (Qwen-7B)
- Evaluation of the personalization task relies on LLM judges.
- On verifiable tasks (GSM etc.), the reason that MIPO outperforms RLVR, which is trained on ground-truth, requires further explanation and an ablation study.

---

> ### Author Rebuttal · Authors · 2026-03-31
>
> We thank reviewer for their questions! Based on your ideas, we conducted additional experiments to address (1) the impact of multiple negative samples per chosen response; (2) scaled up experiments with Llama-3.3-70B-Instruct; and (3) performance of RLVR with outcome rewards.
>
> ### 1. Re “What is the affection when multiple negatives per chosen response are used?”
> Following this suggestion, we compared $N=1$ negative samples to $N \in \{3, 5, 10\}$, where each rejected sample $y_r \sim p(y|x, c')$ is paired with a single chosen response $y_c \sim p(y|x, c)$ on the MultiBench (MB) dataset.
> | N | 1 | 3 | 5 | 10 |
> | :--- | :---: | :---: | :---: | :---: |
> | **Ll1ama-B** | 92.9 | 89.3 | 91.7 | **94.1** |
> | **Llama-3B** | 92.9 | **95.2** | **95.2** | 92.9 |
> | **Qwen-1.5B** | 50.0 | 54.8 | 57.1 | **58.3** |
> | **Qwen-3B** | 76.2 | 75.0 | **81.0** | 78.6 |
>
> This is an excellent point, since a larger N would make the lower bound on the mutual information tighter, and we observe that overall performance gains increase with a larger N. We additionally compared 4 different choices of marginal approximation, including missing contexts $c$, $y_r \sim p(y|x)$ (please refer to our response to yayS Section2 for more results).
>
> ### 2. Re "Gains become modest with increased model size":
> We trained meta-llama/Llama-3.3-70B-Instruct using MIPO on the MultiBench dataset with lora rank 32. Then we report the win rate of the MIPO versus the baseline Instruct model across a suite of LLM-judges: GPT-4.1, GPT-5.2, Gemini-2.5-Flash, Gemini-3.1-Flash-Lite-Preview. MIPO is consistently preferred over the 70B baseline, which is also given the same user context during inference. "Missing" and "Random" are two versions of negative sampling used during MIPO data construction.
>
> | Win rate | Missing (MIPO) | 70B-Instruct | Random (MIPO) | 70B-Instruct |
> | :--- | :---: | :---: | :---: | :---: |
> | **Gemini-3.1** | **60.7** | 39.3 | **60.7** | 39.3 |
> | **Gemini-2.5** | **61.9** | 38.1 | **58.9** | 41.1 |
> | **GPT 5.2** | **54.2** | 45.8 | **61.1** | 38.9 |
> | **GPT 4.1** | **60.1** | 39.9 | **57.1** | 42.9 |
>
> For LLM-judge evaluation, we presented candidate responses as `(response_a, response_b)`, vice versa, along with the personalization rubric to select the preferred response. While the gains are more substantial for smaller models, we still observe improvements with larger models, even 70B-Instruct, which is already trained for instruction following and general response quality.
>
> ### 3. Re “Why does MIPO outperforms RLVR methods on verifiable tasks?”:
> This is an excellent point. Even with access to ground-truth steps and answers, the reward model may misreward the generator. For example, the learned Qwen 1.5B policy outputs partial reasoning without ever providing the final answer, e.g., ```"Step 1: Identify the total number of pizzas which is given as **4**."```
>
> The prompt we provided to the process reward model is:
> ```
> "You are a helpful math assistant, that evaluates another assistant's response based on how accurately the response solves the given problem. Here's the correct response: [Step]\n#### [Answer]\n\nGive a score between 1 and 5 based on partial correctness. 1 if the reasoning is poor, and 5 if the reasoning is correct and the answer is correct. \n\n[Question]"
> ```
> Even though the instruction explicitly states 5 if both reasoning and answer are correct, the reward models may provide noisy signals for partial and incorrect responses, thereby derailing RLVR performance. To improve this aspect of the results, we replaced the above reward with exact string matching suggested by bEwP (e.g., extracting the final answer and giving 1 if it matches the target answer). We found that this increased performance overall and particularly boosted llama1B on GSM8k (see below table), but MIPO still occasionally outperforms RLVR.
>
> | Model | GSM (Process Reward) | GSM (Binary Reward) | GSM (MIPO) | SVAMP (Process Reward) | SVAMP (Binary Reward) | SVAMP (MIPO) |
> | :--- | :---: | :---: | :---: | :---: | :---: | :---: |
> | **Llama-1B** | 10.7 | 28.0 | **29.5** | 52.7 | 57.3 | **60.1** |
> | **Llama-3B** | **71.7** | 69.0 | 70.17| 79.1 | **79.3** | 78.2 |
> | **Qwen-1.5B** | 62.5 | 65.0 | **71** | **83.0** | 80.3 | 81.7 |
> | **Qwen-3B** | 82.8 | 74.5 | **89.2** | 90.2 | 90.7 | **91.3** |
>
> One potential limitation with sparse, binary rewards based on exact string match is that the following rollout by Qwen received 0 even though the reasoning is correct:
> ```
> ... The correct number of pieces made from cashmere is:
> c = 6 (pieces)
> To find out the number of pieces made of a blend, we subtract the number of pieces made from cashmere from the total number:
> Rational:
> 13 (total) - 6 (cashmere) = 7
> To convert this to a mixed number:
> 7 = 2 (whole) + 5 (fraction)
>
> Ground truth:  2
> Extracted answer: NA
> ```
> Even though Qwen eventually achieves an avg. 0.82+ reward (max: 1) after 600 training steps with RLVR, it performs slightly worse on the test set.

---

> > ### Author Rebuttal · Reviewer_r2iM · 2026-04-07
> >
> > Thank you for the response and supplementary experiments. However, some concerns remain.
> > For W1, while the 70B results demonstrate consistent improvements, they do not address the originally raised concern about limited gains at the 7B scale. For W2, the authors partially mitigate position bias through response-order permutation, but verbosity bias and self-consistency bias remain unaddressed. More fundamentally, it is still unclear why LLM judges — which themselves lack ground truth for personalization — are reliable evaluators for this task. For W3, the additional ablation with binary rewards is well-motivated and satisfactorily addresses the question.
> > I keep my score.

---

> > > ### Author Response · Authors · 2026-04-08
> > >
> > > Thank you for your valuable feedback. We hope the clarifications below can help address your remaining concerns:
> > >
> > > ### Re "limited gains at the 7B scale.":
> > > You’re right that gains do vary by model size. We see 10-90% increase for 1/1.5B, 6-18\% for 3B, and 3-5\% for 7B compared to personalized prompting baselines, and similarly, for math and reasoning benchmarks, we observe 1-3.6\% improvements for bigger models compared to ~18\% for Llama 1B. We hypothesize these differences may indicate how heavily fine-tuned a particular model is for a particular domain. For example, we suspect Qwen2.5-7B-Instruct has been heavily fine-tuned to perform well on e.g. GSM8k. We would like to point out that across the 40 different model & benchmark combinations we ran, MIPO rarely decreases performance. Therefore it can be seen as a sometimes highly beneficial, sometimes neutral model enhancement step (Our motivating question for this work was, if we think back to the literature on computer vision, e.g. Imagenet by Krizhevsky et al., 2012, data augmentation approaches like translation and rotation improved learning without the use of any external supervision. Can we develop an analogous approach for LLMs, creating reliable learning signals intrinsic to the problem, without relying on human data or verifiers?).
> > >
> > > ### Re "More fundamentally it is still unclear why LLM judges — which themselves lack ground truth for personalization — are reliable evaluators for this task":
> > > This is an excellent point, and we agree that it is important to validate the reliability of LLM judges. Thus we evaluated the LLM judges used in our paper (Qwen 14B-Instruct and Llama 8B-Instruct) on the ground-truth user preferences from the Community Alignment (CA) and PRISM datasets. Llama 8B achieved 92\% preference prediction accuracy on both datasets, and Qwen 14B achieved 93.5\% and 92.2\%, respectively (please see Section 5.2 Evaluation for further details.)
> > >
> > > Additionally, our evaluations span both verifiable and non-verifiable domains. For verifiable tasks, performance is measured using exact answer matching without LLM judges. For non-verifiable domains, while we acknowledge the limitations of LLM-based evaluation (e.g., verbosity bias and self-consistency bias as mentioned) and are happy to further discuss them in the Limitations section, LLM-based automatic evaluation is widely adopted for cost efficiency and replicability. For example, AlpacaEval 2.0 (https://github.com/tatsu-lab/alpaca_eval) uses GPT-4 as a judge and demonstrates strong correlation with human evaluations from ChatBot Arena (Spearman correlation 0.93–0.98 based on 20K annotations). We will provide additional qualitative examples to further showcase MIPO's advantages.
> > >
> > > ### Re "For W3, the additional ablation with binary rewards is well-motivated and satisfactorily addresses the question.":
> > > We are glad this helped clarify your question. We will add the binary reward results to our Math and MCQ table. Thank you again for your time and valuable review to help us improve our work.

---

### Official Review · Reviewer_bEwP · 2026-03-12

**Soundness:** 4
**Presentation:** 3
**Significance:** 3
**Originality:** 2
**Overall Recommendation:** 5
**Confidence:** 4

**Summary:**

This paper introduces MIPO (Mutual Information-based Preference Optimization), a self-training method that constructs DPO preference pairs without human labels or external verifiers. The key idea is to use a model's own response to the correct prompt as the positive sample, and a response generated from a random or incomplete prompt as the negative. The authors establish a theoretical connection between their training objective and InfoNCE, showing that MIPO implicitly maximizes pointwise mutual information between prompts and responses under the reference policy. Empirically, MIPO is evaluated on personalization tasks (PRISM, Community Alignment, Multi-Bench) and general benchmarks (GSM8k, MMLU, ARC), showing strong performance especially in sub-10B models and low-data regimes.

**Compliance With Llm Reviewing Policy:**

Affirmed.

**Key Questions For Authors:**

1. Where’s the DPO baseline in the main paper? In the appendix D the author stated “DPO is trained using the DPO implementation of OpenRLHF” yet in the main paper, no baseline is DPO.

2. One potential issue in MMLU experiments is that only 200 subsets are evaluated, also I don’t think this dataset is a well suited choice here as the main idea is just to train instruction following capability. Since the improvement we can observe from MMLU with external reasoning is mainly irrelevant. Maybe remove this or substitute with IF-Eval. My suggestion is try MMLU-Redux if you’re resource limited, at least the subset is cleaned and contains less noise.

3. Some issues with GSM8k numbers in RLVR vs SFT in all models, it seems except Llama-3.2-3B-Instruct, all models seem to perform worse in RLVR  than SFT. Either the reward model is not performing well or the author should just use string comparison instead ( which is reasoning for MCQA, GSM8k tasks )

4. Did you do hyperparams search on all baselines methods as well?

**Limitations:**

I would suggest the author do actually discuss their limitations in the limitation section 7, cause right now no limitation was mentioned in that section. One potential limitation might be since the supervised signal came from the model and the contrast signal between prompts, if the model doesn’t have the ability to generate high enough contrast signal ( for example the generated response itself is very long in nature, ie 10k tokens ) or the prompt are so similar among each other that sampling other responses as negative wouldn’t result in useful signals.

For example: MIPO only maximizes pointwise MI under the reference policy (π_ref), not the learned policy (π_θ). The paper itself acknowledges that true MI maximization would require an online RL loop (Eq. 11), yet this fundamental gap between the method's theoretical motivation and what it actually optimizes is not discussed in the limitation section. This is a meaningful theoretical weakness that deserves explicit treatment.

**Strengths And Weaknesses:**

Strengths:
1. The theoretical grounding connecting the DPO loss to InfoNCE is sound and well-motivated. The method is clean and conceptually simple.

2. Strong empirical results in personalization tasks, with 4–38% improvements over personalized prompting baselines, and particularly meaningful gains for smaller 1–1.5B models even when self-generated data is suboptimal.

3. The low-data regime performance and label-free nature make MIPO practically appealing for personalization and other non-verifiable tasks where reward signals are expensive.

Weakness:

1. **Critical** : One missing experimentation is the scaling of high quality and quantity of data over MIPO and other supervised learning method ( SFT, DPO ), cause its not well studies what was the limits of such unsupervised learning method would achieve or the ceiling it would hit. Without proper external signals, MIPO is essentially a self improving LLM which eventually would plateau in performance.

2. The MI lower bound from InfoNCE tightens as the number of negatives N increases. MIPO uses N=2 (one positive, one negative), which is the loosest possible bound. There is no ablation on whether using more negatives per prompt meaningfully improves performance, which is an obvious experiment given the theoretical framing.


      a.  In practice, MIPO generates one chosen response per prompt and reshuffles them as negatives, rather than truly sampling i.i.d. from the marginal p(y). The paper acknowledges this creates "a gap between theory and practice," but doesn't empirically quantify its effect. Correlated positives and negatives could weaken the contrastive signal, however there’s no such analysis experiments included.

---

> ### Author Rebuttal · Authors · 2026-03-31
>
> We thank reviewer for their feedback. Based on your ideas, we added ablations studying: (1) the impact of negative N; (2) reshuffling v.s. Iid sampling; (3) comparison of MI under the base policy and the learned policy; (4) RLVR using exact string matching; and (5) scaling to 70B Instruct model.
>
> ### 1) Re “MIPO uses N=2 (one positive, one negative), which is the loosest possible bound.”
> To address this point, we compared $N=1$ negative samples to $N \in \{3, 5, 10\}$, where each rejected sample $y_r \sim p(y|x, c')$ (iid sampling) is paired with a single chosen response $y_c \sim p(y|x, c)$ on the MultiBench (MB) dataset, and observe that performance generally increases with N, as the theory suggests.
>
> | N | 1 | 3 | 5 | 10 |
> | :--- | :---: | :---: | :---: | :---: |
> | **Ll1B** | 92.9 | 89.3 | 91.7 | **94.1** |
> | **Ll3B** | 92.9 | **95.2** | **95.2** | 92.9 |
> | **Qw1.5B** | 50.0 | 54.8 | 57.1 | **58.3** |
> | **Qw3B** | 76.2 | 75.0 | **81.0** | 78.6 |
>
> ### 2) Re “Correlated positives and negatives could weaken the contrastive signal”:
> This is a great point, and the following table compares reshuffling to i.i.d. sampling of the rejected responses on MultiBench (MB) and Community Alignment (CA). Note that reshuffling was a practical implementation choice to further reduce data generation costs (but not a necessary element of MIPO) and to eliminate the potential concern that MIPO's relative performance gains compared to SFT are due to the bigger training data size.
>
> | Model | CA "Random" | CA Reshuffling | MB "Random" | MB Reshuffling |
> | :--- | :---: | :---: | :---: | :---: |
> | **Llama1B**  | **88.5** | 87.0  | **92.9** | 89.7 |
> | **Llama3B** | **90.5** | 88.0  | 92.9 | **94.8** |
> | **Qwen1.5B**  | **78.0** | 68.0  | 50.0 | **56.0** |
> | **Qwen3B**  | 78.0 | **80.3**  | **76.2** | 70.6 |
>
> ### 3) Re “MIPO only maximizes pointwise MI under the reference policy (π_ref), not the learned policy (π_θ)”:
> Yes, this is true, as we also acknowledge in the paper. We hypothesize that this could actually be a strength, since optimizing the mutual information under the learned policy could incentivize reward hacking via manipulating the policy’s output probabilities, without changing the underlying action distributions. To provide support for this, we compared to directly maximizing the pointwise MI under the learned policy and InfoNCE. Please refer to TiWN Section3 for full results.
>
> | Model | CA (MIPO) | CA (Online MI) | CA (InfoNCE) | MB (MIPO) | MB (Online MI) | MB (InfoNCE) |
> | :--- | :---: | :---: | :---: | :---: | :---: | :---: |
> | **Llama3B** | **90.3** | 72.7 | 83.7 | **94.8** | 90.1 | 93.7 |
> | **Qwen3B** | **80.3** | 72.7 | 70.0 | **76.2** | 66.7 | 73.0 |
>
> ### 4) Re “author should just use string comparison” in RLVR.
> We incorporated your advice and experimented with using more sparse, binary rewards based on exact string match with a target answer. This increased performance overall, and particularly boosted llama1B on GSM (see the below table), but MIPO still occasionally outperforms.
>
> | Model | GSM (Process Reward) | GSM (Binary Reward) | SVAMP (Process Reward) | SVAMP (Binary Reward) |
> | :--- | :---: | :---: | :---: | :---: |
> | **Ll1B** | 10.67 | **28.00** | 52.66 | **57.33** |
> | **Ll3B** | **71.67** | 69.00 | 79.11 | **79.33** |
> | **Qw1.5B** | 62.50 | **65.00** | **83.00** | 80.33 |
> | **Qw3B** | **82.83** | 74.50 | 90.22 | **90.67** |
>
> ### 5) Experiments with larger models
> We also trained Llama-3.3-70B-Instruct on MultiBench with lora rank 32. Then we report the win rate of the MIPO versus the baseline 70B-Instruct model across 4 frontier LLM-judges. MIPO is consistently preferred over the 70B baseline, which is given the same user context and query as MIPO.
>
> | Win rate | MIPO | 70B-Instruct |
> | :--- | :---: | :---: |
> | **Gemini-3.1-Flash-Lite-Preview** | **60.7** | 39.3 |
> | **Gemini-2.5**  | **58.9** | 41.1 |
> | **GPT 5.2**  | **61.1** | 38.9 |
> | **GPT 4.1**  | **57.1** | 42.9 |
>
> ### 6) Re "One missing experimentation is the scaling of high quality and quantity of data"
> For the ablations with N negatives, we observe that accuracy on training curves eventually plateau (e.g., llama1& 3b and qwen1.5& 3b with a learning rate 1e-6 achieve average acc. of 98% around 6k steps on multibench). We'll include the learning curves in the appendix.
>
> ### 7) Re "Where’s the DPO baseline in the main paper?"
> We apologize for the confusion. “DPO”refers to the learning algorithm within MIPO; will correct this typo.
>
> ### 8) Re issues with MMLU:
> Thank you for bringing this to our attention via the MMLU-Redux paper. We will remove MLMU as suggested, and explore alternative datasets.
>
> ### 9) Re hyperparameters
> For SFT and InfoNCE, we swept over learning rates {1e-6, 1e-7, 5e-7}. PPO (RLVR and RLAIF) was implemented with an actor learning rate 5e-7 (Appendix D).
>
> ### 10) Re "if the model doesn’t have the ability to generate high enough contrast signal… as negatives wouldn’t result in useful signals."
> We will add to Limitations.

---

> > ### Author Rebuttal · Reviewer_bEwP · 2026-04-03
> >
> > I enjoy reading the additional info the authors provide ( specifically 2 ), I would maintain my current score as its currently already "accept". I wouldn't increase the score but I would be willing to argue with other reviewers for at least a weak accept score in the later discussion period.

---

> > > ### Author Response · Authors · 2026-04-07
> > >
> > > Thank you for your continued support for our work. Following your suggestion, we expanded reshuffling ablations across both personalization and reasoning domains with more seeds, and provide a much more thorough analysis below. We find that **our expanded results match your understanding that reshuffling may "weaken contrastive signal"**, which also aligns with our theory.
> > >
> > > Here’s the updated table with community alignment (CA), multifaceted bench (MB), ARC easy and challenge (MCQ), and GSM (Math) with 3 seeds (we report mean and standard deviation in the parentheses):
> > >
> > > | Model | CA (Personalization) | MB (Personalization)  | ARC easy (MCQ) | ARC challenge (MCQ) | GSM8 (Math) |
> > > |:------|:--|:--|:-----|:-----|:---|
> > > | **Llama1B** | 78 | 79.76 | 44 | 33.2 | 22 |
> > > |  -- Reshuffling | 88.33 (2.02) | 84.52 (2.38) | 51.87 (2.01) | 39.53 (3.11) | **29.5 (2.29)** |
> > > |  -- Random (iid) | 87 (5.57) | 89.68 (1.82) | **53.47 (2.80)** | **41.13 (2.25)** | 27.33 (1.26) |
> > > |  -- Missing contexts (iid) | **93.67 (1.26)** | **93.26 (0.69)** | - | - | - |
> > > | **Llama3B** | 78 | 83.33 | 80.4 | 68.6 | 71 |
> > > |  -- Reshuffling  | 86.83 (1.89) | 91.27 (1.82) | 85.26 (0.58) | 70.93 (0.7) | 70.17 (2.02) |
> > > |  -- Random (iid) | 88 (2.29) | **94.84 (1.82)** | **85.4 (0.69)** | **72.67 (1.81)** | **72.17 (0.29)** |
> > > |  -- Missing contexts (iid) | **90.33 (2.08)** | 94.45 (2.48) | - | - | - |
> > > | **Qwen1.5B**  | 63.5 | 39.29 | 79 | 63.6 | 65.5 |
> > > |  -- Reshuffling  | 71.83 (1.53) | 40.87 (3.44) | **82.27 (0.81)** | 65.93 (1.7) | **71 (1.80)** |
> > > |  -- Random (iid) | 68 (1) | 55.95 (3.57) | 81.87 (0.31) | **66.53 (1.10)** | 70.33 (1.15) |
> > > |  -- Missing contexts (iid) | **78.83 (2.84)** | **74.6 (3)** | - | - | - |
> > > | **Qwen3B**  | 76 | 63.1 | 92 | 79.4 | 84.5 |
> > > |  -- Reshuffling  | 76.17 (1.76) | 65.87 (4.51) | 90.80 (0.35) | 80.13 (2.01) | **89.17 (3.75)** |
> > > |  -- Random (iid) | **80.33 (1.76)** | 70.63 (2.48) | **91.13 (0.42)** | **80.8 (0.8)** | 86.67 (2.25) |
> > > |  -- Missing contexts (iid) | 79.83 (0.29) | **74.6 (0.69)** | - | - | - |
> > >
> > > These additional results show that iid sampling is especially useful for personalization, because iid sampling with missing contexts is an exact Monte Carlo approximation of sampling from $p_\text{ref}(y|x)$. For personalization, our theory requires that the rejected samples are generated from $p_\text{ref}(y|x)$ and the chosen responses are generated from $p_\text{ref}(y|x,c)$ to maximize the conditional mutual information $I_{\text{ref}}(y; c | x)$ under the reference policy, so sampling rejected responses without user-specific contexts is most aligned with the theory. In contrast, random contexts (iid) makes an additional approximation by doing $p_{\text{ref}}(y|x) = \mathbb E_{c’} [p_\text{ref}(y|x, c’)]$ and is often outperformed by missing contexts $p_\text{ref}(y|x)$. And our comparisons of reshuffling and missing contexts (iid) show that the latter achieves better performance. Please note that during the rebuttal, we had mistakenly said that our original results used reshuffling for personalization, but in fact we originally did iid sampling for personalization, and reshuffling was used only for the reasoning and math datasets. This is because, for the personalization dataset, prompts $x$'s are mostly non-overlapping, so combining $x$ with a new, random context $c’$ requires sampling a new rejected response for the particular $(x, c')$. Likewise, generating the response $y$ from $x$ alone with missing contexts requires sampling a new rejected response. Correcting this, we have now added the true reshuffling results for personalization, where the rejected samples are responses to different prompts and contexts, and found that it performs worse, as you expected.
> > >
> > > For math and reasoning benchmarks, our paper makes a practical suggestion of reshuffling the chosen response to a different prompt $x’$ as the rejected response to the prompt $x$, but the theory requires sampling from $p(y) = E_{x’} [p(y|x’)]$. This second option is now added as “random (iid)” for the math and reasoning benchmarks. (Please note that “random (iid)” for math and reasoning refer to random prompts $p_\text{ref}(y|x’)$, since these tasks do not have additional user-contexts.) With these additional results, we also observe a slight boost in performance with iid sampling compared to reshuffling, which again aligns with your understanding and our theory. Interestingly, in GSM8k, we see similar results between reshuffling and iid sampling, so the magnitude of effects varies by empirical domains, but the general trends show that iid sampling is superior across both personalization, MCQ, and math.
> > >
> > > Thank you for giving us the chance to clarify and elaborate on our ablation results. We hope this helps address any remaining questions, and we’re happy to keep incorporating your suggestions into our work.

---

### Official Review · Reviewer_TiwN · 2026-03-16

**Soundness:** 3
**Presentation:** 2
**Significance:** 2
**Originality:** 2
**Overall Recommendation:** 3
**Confidence:** 3

**Summary:**

This paper proposes MIPO, a DPO-based self-training method that forms contrastive pairs by pairing outputs from the correct prompt/context with outputs from random or incomplete prompt/context pairings, and motivates this through a pointwise mutual-information view under the reference policy. The empirical results, especially on personalization, are interesting and the method is simple to implement.

**Compliance With Llm Reviewing Policy:**

Affirmed.

**Final Justification:**

Thank you for the detailed clarifications and additional experiments—this is a helpful and thorough rebuttal.

On dependence on stronger models, I appreciate the clarification that GPT-4o is only used for preprocessing and that contexts are shared across baselines. The analogy to system prompts in deployed systems is helpful. That said, I still feel the “self-supervised” framing could be clarified more carefully in the paper, especially for the personalization setting where such contexts are externally constructed.

On MI under reference vs. learned policy, I appreciate the additional comparisons and discussion. I view this concern primarily as a presentation issue, and I agree that clarifying the role of the MI interpretation and its variants will strengthen the paper.

On larger model results, thank you for including additional experiments and for agreeing to incorporate rubric/benchmark evaluations in the main text. I think this will significantly improve consistency and interpretability.

On the i.i.d. vs. reshuffling discussion, the expanded experiments and clarification are particularly helpful. It is useful to see that i.i.d. sampling generally aligns better with the theoretical objective and can outperform reshuffling, while reshuffling remains a practical tradeoff. This clarification strengthens the paper. At the same time, I still think it would be valuable to explicitly discuss this theory–practice gap and provide intuition for when each variant is expected to perform better.

Overall, I appreciate the authors’ efforts to address the concerns. While the rebuttal improves clarity and strengthens the empirical case, my remaining concerns are mainly about presentation and framing. I therefore intend to keep my current rating.

**Key Questions For Authors:**

How much the reshuffling approximation matters?

**Limitations:**

Judge-based evaluation: Most evaluations rely on LLM judges, whose strict accuracy is imperfect, leaving some uncertainty about the robustness of the reported gains.

Limited gains for larger models: The improvements are much smaller for larger models, suggesting the method mainly benefits weaker models that struggle to use contextual information.

Lack of failure analysis: The paper provides little analysis of failure cases, making it unclear when the contrastive prompt–response pairing strategy may not work or could degrade performance.

**Strengths And Weaknesses:**

**Strengths**
- The experimental design about the comparison between SFT and MIPO is clear, done on the same self-generated chosen responses. That isolates the value of the contrastive signal instead of conflating it with extra data. The setup is also broad enough to be meaningful: five models, multiple personalization datasets, and multiple benchmark families.
- Experimental results are strong, MIPO beats personalized prompting across all five models, with especially large gains for smaller models, and the paper also reports that the same contrastive idea helps on benchmark tasks.

**Weakness**
- The paper may overstate the label-free part. The presentation emphasizes that the method does not require human labels or verifiers and only needs prompts plus the model being trained, but for the key personalization datasets the pipeline uses GPT-4o to infer/verbalize user preferences into contexts, and the appendix provides the explicit GPT-4o prompt used for this preprocessing. In addition, personalization is evaluated with larger judge models. This makes the empirical setup look closer to supervision transferred from stronger models than to purely intrinsic self-improvement, so the framing feels overstated.

- The theory-to-method connection is also stated too strongly, at least highlighted as a key message in title. The formal argument assumes negatives sampled from the marginal distribution, but the paper later clarifies that the optimized quantity is pointwise mutual information under the reference policy, not mutual information under the learned policy. The practical method then further replaces marginal sampling with a one-sample Monte Carlo approximation and reshuffling, and the paper explicitly acknowledges a gap between theory and practice. So the mutual-information interpretation is a useful intuition, but not a tight characterization of the implemented algorithm.

The two concerns are related: once stronger-model preprocessing is involved and the MI connection is only approximate, the paper should be more careful about claiming a clean, purely intrinsic self-improvement mechanism.

---

> ### Author Rebuttal · Authors · 2026-03-31
>
> Thank you for your questions! Based on your feedback, we conducted ablations to address: (1) the impact of reshuffling; (2) adding N negatives; (3) comparison between optimizing the MI of the reference policy versus the learned policy; and (4) experiments with meta-llama/Llama-3.3-70B-Instruct.
>
> ### (1) Re “How much the reshuffling approximation matters?”:
> The following table compares reshuffling to i.i.d. sampling $y_r \sim p(y|x, c’)$ with a random context $c’$  on MultiBench (MB) and Community Alignment (CA):
>
> | Model | CA "Random" | CA Reshuffling | MB "Random" | MB Reshuffling |
> | :--- | :---: | :---: | :---: | :---: |
> | **Llama1B**  | **88.5** | 87.0  | **92.9** | 89.7 |
> | **Llama3B** | **90.5** | 88.0  | 92.9 | **94.8** |
> | **Qwen1.5B**  | **78.0** | 68.0  | 50.0 | **56.0** |
> | **Qwen3B**  | 78.0 | **80.3**  | **76.2** | 70.6 |
>
> While reshuffling the chosen responses can create noisy negative signals, empirically, we observe it to be effective while reducing data generation costs, but iid sampling is indeed closer to our theory.
>
> ### (2) Re “The practical method further replaces marginal sampling with a one-sample MC...":
> We added experiments sweeping $N= \{1, 3, 5, 10 \}$ negatives, where rejected samples $y_r \sim p(y|x, c')$ are generated with random contexts $c'$ on MultiBench. We find performance generally improves with larger N, as it tightens the MI lower bound.
>
> | N | 1 | 3 | 5 | 10 |
> | :--- | :---: | :---: | :---: | :---: |
> | **Ll1B** | 92.9 | 89.3 | 91.7 | **94.1** |
> | **Ll3B** | 92.9 | **95.2** | **95.2** | 92.9 |
> | **Qw1.5B** | 50.0 | 54.8 | 57.1 | **58.3** |
> | **Qw3B** | 76.2 | 75.0 | **81.0** | 78.6 |
>
> ### (3) Re “... the optimized quantity is pointwise mutual information under the reference policy, not mutual information under the learned policy”:
> Yes, this is true, and we acknowledge this point in the paper. We hypothesize that this could actually be a strength, since the reference policy is a reliable model of language. Optimizing the MI under the learned policy could incentivize reward hacking via manipulating the policy’s output probabilities, without changing the underlying action distributions. To provide support for this, we directly compared to using online RL to maximize MI under the learned policy (“online MI” in the below table) and InfoNCE. Since optimizing MI under the learned policy requires an online learning algorithm, we used PPO with a pointwise MI-reward $\log \pi_\theta(y|x,c) - \log \pi_\theta(y|x)$. MIPO consistently outperforms online MI-PPO and InfoNCE.
>
> | Model | CA (MIPO) | CA (Online MI) | CA (InfoNCE) | MB (MIPO) | MB (Online MI) | MB (InfoNCE) |
> | :--- | :---: | :---: | :---: | :---: | :---: | :---: |
> | **Llama1B** | **93.6** | 80.2 | 89.5 | **93.3** | 82.2 | 91.7 |
> | **Llama3B** | **90.3** | 72.7 | 83.7 | **94.8** | 90.1 | 93.7 |
> | **Qwen1.5B** | **78.8** | 39.7 | 58.8 | **74.6** | 36.9 | 59.5 |
> | **Qwen3B** | **80.3** | 72.7 | 70.0 | **76.2** | 66.7 | 73.0 |
>
> ### (4) Re “Limited gains for larger models”:
> We trained meta-llama/Llama-3.3-70B-Instruct using MIPO on MultiBench with lora rank 32. Then we report the win rate of the MIPO (using missing and random contexts) versus the baseline 70B-Instruct model across LLM-judges: GPT-4.1, GPT-5.2, Gemini-2.5-Flash, Gemini-3.1-Flash-Lite-Preview. MIPO is consistently preferred over the 70B baseline, which is also given the same user context.
> | Win rate | Missing (MIPO) | 70B-Instruct | Random (MIPO) | 70B-Instruct |
> | :--- | :---: | :---: | :---: | :---: |
> | **Gemini-3.1** | **60.7** | 39.3 | **60.7** | 39.3 |
> | **Gemini-2.5** | **61.9** | 38.1 | **58.9** | 41.1 |
> | **GPT 5.2** | **54.2** | 45.8 | **61.1** | 38.9 |
> | **GPT 4.1** | **60.1** | 39.9 | **57.1** | 42.9 |
>
> For LLM-judge evaluation, we presented candidate responses as `(response_a, response_b)`, vice versa, along with the personalization rubric.
>
> ### (5) Re “empirical setup look closer to supervision transferred from stronger models”:
> We present MIPO as a self-supervised method in that no stronger models are used to *generate responses* – unlike standard strong-to-weak transfer learning where responses are generated or revised by a stronger model. However it assumes datasets of queries and user-contexts (see the example below). Since existing real-user datasets lack explicit ground-truth user contexts, we utilized GPT-4o to infer such contexts. Importantly, these contexts were also given to all the baselines, except for the default generation. We're happy to clarify this in Limitations.
>
> Below is an example context from the PRISM dataset that is provided to all methods alongside a user's query:
> ```
> 'Favor short and direct explanations over detailed ones, even when the topic is complex Show empathy and acknowledge the difficulties faced by individuals or groups in challenging situations. Communicate in a simple and straightforward manner without unnecessary elaboration.'
> ```
> ### (6) Re "Judge-based evaluation":
> We will discuss this in Limitations.

---

> > ### Author Rebuttal · Reviewer_TiwN · 2026-04-03
> >
> > Some of the comments are on the presentation side, not soundness. These are currently not fully resolved from the rebuttal. I would encourage to clarify them explicitly in the paper. These are the following:
> >
> > - **Dependence on stronger models** The clarification that GPT-4o is only used for preprocessing is helpful. However, it still seems that the “self-supervised” framing is somewhat overstated, since the setup depends on external model-derived contexts.
> >
> > - **MI under reference vs. learned policy** The comparison to online MI optimization is helpful and supports the empirical choice of the reference-policy formulation. Similarly, my comments earlier is more on the presentation sight, in that "how central is the MI interpretation" to this paper.
> >
> > On the **larger model resutls**: Thanks the author for providing results on other comments too, the "larger model results" is helpful. However, I would like to suggest the authors add rubric/benchmark tasks in the manuscript. Judge-based evaluation can be very judge-biased, especially for large models (where the model's capabilities are strong and making the decision boundary vauge). That being said, at the rebuttal period, I understand the choice for providing "win-rate" results.
> >
> > The difference provided in the **iid vs. reshuffling** is actually meaningful to understand the theory. The results show that reshuffling can either outperform or underperform i.i.d. sampling depending on the setting. It is unclear how the theoretically grounded method relates to the empirical method in practice, leaving some questions about the theoretical framing. This is worth discussing and some relevant insights will make the paper more interesting.
> >
> > The comments are currently partially resolved, particularly from the presentation perspective, I intend to keep my current rating.

---

> > > ### Author Response · Authors · 2026-04-07
> > >
> > > Thank you for giving us the opportunity to clarify. We will fix the presentation based on your suggestion as follows:
> > >
> > > ### Re “dependence on stronger models”:
> > > Please note that the performance of MIPO on general reasoning and math domains does not require strong model-derived contexts and therefore involves no additional supervision. For personalization, we needed to obtain a user context c, and this is not easily available in many datasets, which is why we created such c’s for the datasets as a preprocessing step, and made this information available to all baseline methods. We believe this setup is analogous to industry dialog models where we assume that the user’s preferences or context are derived through some heavily curated process, then made explicit in the system prompt. Thus we evaluate only the policy’s ability to adapt to these user-specific details in its response.
> > >
> > > An alternative to using user contexts would be to use past conversations or self-stated preferences (if available) as “c.” However, users’ self-stated preferences are typically not available in real-world datasets unless they are collected through separate surveys. Moreover, even state-of-the-art reward models conditioned on users’ self-stated preferences or past conversations achieve only 60–62% accuracy in preference prediction (arxiv.org/abs/2507.13579) which may affect the reliability of the reward model when used to evaluate personalized responses. We will discuss the limitations of existing personalized benchmarks and how they affect our evaluation in the Limitations. We can also rename the section “Personalized instruction following” to better denote the task of following instructions based on generated user contexts. If you have further suggestions for presentation, we're happy to incorporate them.
> > >
> > > ### Re “MI under reference vs. learned policy”:
> > > This is a great point, we will clarify that we tested multiple methods for maximizing mutual information, either under the learned policy, or with respect to the base policy. It is interesting that using the base policy offers superior performance, and we will add a discussion of this and our hypothesized reasoning which we wrote in the rebuttal above.
> > >
> > > ### Re “larger model results”:
> > > We will include the rubric/benchmarks in the main text based on your suggestion.
> > >
> > > ### Re “difference in iid vs. reshuffling”:
> > > This is an excellent point and following your suggestion, we expanded reshuffling ablations across both personalization and reasoning domains with more seeds, and provide a much more thorough analysis below. Please review our Reply Rebuttal to bEwP for the updated table across 5 domains with 3 seeds.
> > >
> > > **Our expanded results match your intuition that iid sampling often outperforms reshuffling**, especially for personalization, because iid sampling is an exact Monte Carlo approximation of sampling from $p_\text{ref}(y|x)$. For personalization, our theory requires that the rejected samples are generated from $p_\text{ref}(y|x)$ and the chosen responses are generated from $p_\text{ref}(y|x,c)$ to maximize the conditional mutual information $I_\text{ref}(y;c|x)$ under the base policy, so sampling rejected responses without user-specific contexts is aligned with the theory. In contrast, random contexts (iid) makes an additional approximation by doing $p_\text{ref}(y|x) = \mathbb E_{c’} [p_\text{ref}(y|x, c’)]$ and is often outperformed by missing contexts. Please note that during the rebuttal, we had mistakenly said that our original results used reshuffling for personalization, but in fact we originally did iid sampling for personalization, and reshuffling was used only for the reasoning and math datasets. This is because, for the personalization dataset, each x is unique, so combining x with a new, random context c’, or generating the response y from x alone requires sampling a new rejected response. Correcting this, we have now added the true reshuffling results for personalization, where the rejected responses are both generated from misspecified contexts and prompts, and found that it performs worse, as you expected. For math and reasoning benchmarks, we also observe a slight boost in performance with iid sampling compared to reshuffling (which aligns with your understanding) though the magnitude of effects vary by empirical domains.
> > >
> > > Other ablations also show that the choice of negatives does matter. One exception to contrastive pair generation that would break MIPO’s theory is, if the rejected responses are not under the distribution of the base policy. To test this case, we added an ablation pairing the chosen responses with poorly chosen negatives, i.e. random alphanumeric strings. We believe this is a case with very strong contrastive signals between the chosen and the rejected responses, but breaks the MIPO assumption. Empirically, we observe little to no improvements (please see our response to yayS Sec 3 “Random negative”). Thank you for giving us the chance to elaborate.

---

### Decision · Program_Chairs · 2026-04-30

**Decision:**

Accept (regular)

**Comment:**

This paper proposes MIPO, a simple DPO-based contrastive data construction method motivated by maximizing prompt-response mutual information under the reference policy. Reviewers generally agreed that the method is technically sound and empirically interesting, especially on personalization tasks and for smaller models.

The rebuttal was helpful and materially improved the paper. In particular, the added experiments on multiple negatives, iid vs. reshuffling, MI variants, and larger-model behavior addressed several of the main concerns and strengthened the empirical support.

The remaining concerns are mainly about framing and scope rather than core soundness. The paper should calibrate its “self-supervised / label-free” claims more carefully in the personalization setting, present the MI interpretation with more care given the practical approximations, and better acknowledge the limitations of LLM-judge-based evaluation and the reduced gains at larger model scales.

Overall, I find the contribution sufficiently solid and useful to merit acceptance, though at the weak-accept level rather than as a clear accept.